# From Cluster Assumption to Graph Convolution: Graph-based Semi-Supervised Learning Revisited

## Abstract

Graph-based semi-supervised learning (GSSL) has long been a research focus. Traditional methods are generally shallow learners, based on the cluster assumption. Recently, graph convolutional networks (GCNs) have become the predominant techniques for their promising performance. In this paper, we theoretically discuss the relationship between these two types of methods in a unified optimization framework. One of the most intriguing findings is that, unlike traditional ones, typical GCNs may not effectively incorporate both graph structure and label information at each layer. Motivated by this, we propose three simple but powerful graph convolution methods. The first is a supervised method **OGC** which guides the graph convolution process with labels. The others are two "no-learning" unsupervised methods: **GGC** and its multi-scale version **GGCM**, both aiming to preserve the graph structure information during the convolution process. Finally, we conduct extensive experiments to show the effectiveness of our methods.

## 1 Introduction

Graph-based semi-supervised learning (GSSL) (Zhu, 2005; Chapelle et al., 2009) is one of the most successful paradigms for graph-structured data learning. Traditional GSSL methods are generally shallow[1] learners, based on the cluster assumption (Zhou et al., 2004) that linked nodes tend to have similar labels. In their practical implementation, a supervised label learning model and an unsupervised graph structure learning model will be jointly considered. This joint learning scheme leverages the wealth of huge unlabeled nodes, to obtain significant performance with very few labels.

With the advancement of deep learning, there are now a variety of graph neural networks (GNNs) (Wu et al., 2020) in the GSSL area. The most well-known methods are graph convolutional networks (GCNs) (Zhang et al., 2019) which generalize the idea of convolutional neural networks (CNNs) (LeCun et al., 1995) to learn local, stationary, and compositional representations/embeddings of graphs. Recently, GCNs and their subsequent variants have achieved great success in lots of applications, like social analysis (Kipf & Welling, 2017) and biological analysis (Fout et al., 2017).

Despite their remarkable success, GCNs still badly suffer from over-smoothing (Li et al., 2018), i.e., dramatic performance degradation with increasing the network depth. In contrast, traditional shallow GSSL methods do not have this issue, although these two kinds of methods both progress through the graph in a similar iterative manner. Moreover, a simple baseline C&S (Huang et al., 2021), which combines non-graph neural networks with some traditional shallow GSSL methods, can match or even exceed the performance of state-of-the-art GCNs. These observations call for a rigorous discussion of the relationship between traditional shallow GSSL methods and recently advanced GCNs, as well as how they can complement each other.

In this paper, we bridge these two directions from a joint optimization perspective. In particular, we introduce a unified GSSL optimization framework which contains both a supervised classification loss (for label learning) and an unsupervised Laplacian smoothing loss (for graph structure learning).

---

[1]Following LeCun et al. (2015), we use the terminology "shallow" to mean those simple linear classifiers or any other classifiers which do not involve lots of trainable parameters.

We demonstrate that, compared to traditional shallow GSSL methods, typical GCNs further consider self-loop graphs, potentially employ non-linear operations, possess greater model capacity, but may not guarantee the simultaneous reduction of these two loss terms at each layer.

Motivated by this, we further propose three simple but powerful graph convolution methods. Specifically, the first one is a supervised method named **Optimized Simple Graph Convolution** (**OGC**). When aggregating information from neighbors at each layer, OGC jointly minimizes both the classification and Laplacian smoothing loss terms by introducing an auxiliary **Supervised EmBedding (SEB)** operator. Intuitively, as this aggregation proceeds, nodes incrementally gain more information from further reaches of the graph, while simultaneously benefiting from supervision.

The others are two "no-learning" unsupervised methods: **Graph Structure Preserving Graph Convolution** (**GGC**) and its multi-scale version **GGCM**. Their aims are both to preserve the graph structure during the feature aggregation procedure. As graph convolution only encourages the similarity between linked nodes, we introduce another novel operator, **Inverse Graph Convolution (IGC)**, to preserve the dissimilarity between unlinked nodes. Through jointly conducting these two convolution operations at each iteration, they both can finally preserve the original graph structure in the embedding space, without the need of learning any parameters.

Our contributions are two-fold. First, through the use of a unified GSSL optimization framework, we discover that, in comparison to traditional GSSL methods, the learning process of typical GCNs may not effectively incorporate both the graph structure and label information at each layer. This finding may provide new insights for designing novel GCN-type methods. Second, by introducing two novel graph embedding operators (SEB and IGC), we propose three simple but powerful graph convolution methods; especially, the introduced two novel operators can be used as "plug-ins" to fix the over-smoothing problem for various GCN-type models.

## 2 PRELIMINARIES

Let $\mathcal{G} = (\mathcal{V}, \mathcal{E})$ be a graph with $n$ nodes $v_i \in \mathcal{V}$, and edges $(v_i, v_j) \in \mathcal{E}$. Let $A \in \mathbb{R}^{n \times n}$ denote the adjacency matrix of this graph, and let $D \in \mathbb{R}^{n \times n}$ denote the diagonal degree matrix of $A$. Let $X \in \mathbb{R}^{n \times d}$ denote the node feature/attribute matrix, i.e., each node $v_i$ has a corresponding $d$-dimensional attribute vector $X_i$. Let $Y \in \mathbb{R}^{n \times c}$ denote the true labels of $n$ nodes coming from $c$ classes, i.e., $Y_i$ is the $c$-dimensional label vector of node $v_i$.

### 2.1 GRAPH-BASED SEMI-SUPERVISED LEARNING (GSSL)

Before the era of deep learning, most GSSL methods are shallow. Assuming a subset $\mathcal{V}_L \subset \mathcal{V}$ of nodes are labeled, these methods generally aim to classify the remaining nodes based on the cluster assumption (Zhou et al., 2004; Zhu, 2005) (i.e., linked nodes tend to have similar class labels), by jointly considering a supervised classification loss ($\mathcal{Q}_{sup}$) and an unsupervised Laplacian smoothing loss ($\mathcal{Q}_{smo}$):[2]

$$\min_{f,g} \quad \mathcal{Q} = \mathcal{Q}_{sup} + \mathcal{Q}_{smo} = \sum_{v_i \in \mathcal{V}_L} \ell\left(g(f(X_i)), Y_i\right) + \mathrm{tr}(f(X)^\top L f(X)), \tag{1}$$

where $\ell(\cdot, \cdot)$ is a loss function that measures the discrepancy between the predicted and target value, $\mathrm{tr}(\cdot)$ is the trace operator, $L = D - A$ is the graph Laplacian matrix, $f(\cdot)$ is an embedding learning function that maps the nodes from the original feature space to a new embedding space, and $g(\cdot)$ is a label prediction function that maps the nodes from the embedding space to the label space.

### 2.2 GRAPH CONVOLUTIONAL NETWORKS (GCNs)

GCNs (Zhang et al., 2019) generalize the idea of convolutional neural networks (CNNs) (LeCun et al., 1995) to graph-structured data. One of the most prominent approaches is GCN (Kipf & Welling, 2017). At each layer, GCN performs two basic operations on each node. The first one is graph convolution, i.e., aggregating information for each node from its neighbors:

$$\tilde{D}^{-\frac{1}{2}} \tilde{A} \tilde{D}^{-\frac{1}{2}} H^{(k)}, \tag{2}$$

where $\tilde{A} = A + I_n$ is the adjacency matrix with self-loops, $I_n$ is a $n$-dimensional identity matrix, and $\tilde{D}$ is the degree matrix of $\tilde{A}$. $H^{(k)} \in \mathbb{R}^{n \times d^{(k)}}$ is the matrix of the learned $d^{(k)}$-dimensional node embeddings at the $k$-th layer, with $H^{(0)} = X$.

---

[2]Here, we omit the hyper-parameter between these two loss terms for simplicity.

The second one is feature mapping, i.e., feeding the above convolution results to a fully-connected neural network layer. Combining these two parts together, at the $k$-th layer, the forward-path of GCN can be formulated as: $H^{(k+1)} = \sigma(\tilde{D}^{-\frac{1}{2}} \tilde{A} \tilde{D}^{-\frac{1}{2}} H^{(k)} \Theta^{(k)})$, where $\Theta^{(k)} \in \mathbb{R}^{d^{(k)} \times d^{(k+1)}}$ denotes the trainable weight matrix at the $k$-th layer,[3] and $\sigma(\cdot)$ is a non-linear activation function like Sigmoid and ReLU. For node classification, at the last layer, GCN will adopt a softmax classifier on the final outputs, and use the cross-entropy error over all labeled nodes as the loss function.

Another representative work of GCNs is SGC (Wu et al., 2019) which simplifies GCN through removing nonlinearities and collapsing weight matrices between consecutive layers. In particular, applying a $K$-layer SGC to the feature matrix $X$ corresponds to: $(\tilde{D}^{-\frac{1}{2}} \tilde{A} \tilde{D}^{-\frac{1}{2}})^K X \Theta^*$, where $\Theta^* \in \mathbb{R}^{d \times c}$ is the collapsed weight matrix. Then, for node classification, it trains a multi-class logistic regression classifier with the pre-processed features. Intuitively, SGC first processes the raw node features based on the graph structure in a "no-learning" way, and then trains a classifier with the pre-processed features.

## 3 UNDERSTANDING GCNs FROM AN OPTIMIZATION VIEWPOINT

For easy understanding, we first analyze the simplest GCN-type method SGC (Wu et al., 2019) and then GCN (Kipf & Welling, 2017).

### 3.1 SGC ANALYSIS

Without loss of generality, we still consider a self-loop graph $\tilde{A}$ with its degree matrix $\tilde{D}$, and use $\tilde{L} = \tilde{D} - \tilde{A}$ to denote the Laplacian matrix. Let $U = f(X)$ denote the learned node embedding matrix, i.e., $U$ is a matrix of node embedding vectors.[4] In addition, we further symmetrically normalize $\tilde{L}$ as $\tilde{D}^{-\frac{1}{2}} \tilde{L} \tilde{D}^{-\frac{1}{2}}$, and initialize $U$ with the node attribute matrix $X$ at the beginning, i.e., $U^{(0)} = X$. As such, the objective function in Eq. 1 becomes:

$$\min_{U, g, U^{(0)} = X} \bar{\mathcal{Q}} = \bar{\mathcal{Q}}_{sup} + \bar{\mathcal{Q}}_{smo} = \sum_{v_i \in \mathcal{V}_L} \ell(g(U_i), Y_i) + \text{tr}(U^{\top} \tilde{D}^{-\frac{1}{2}} \tilde{L} \tilde{D}^{-\frac{1}{2}} U), \tag{3}$$

where $\bar{\mathcal{Q}}_{sup}$ is the supervised classification loss, and $\bar{\mathcal{Q}}_{smo}$ is the unsupervised smoothing loss.

From an optimization perspective, we can reformulate the graph convolution (i.e., Eq. 2) used in SGC as the standard gradient descent step:[5] $U^{(k+1)} = U^{(k)} - \frac{1}{2} \frac{\partial \bar{\mathcal{Q}}_{smo}}{\partial U^{(k)}}$. In addition, the smoothing loss $\bar{\mathcal{Q}}_{smo}$ in Eq. 3 can be reduced with an adjustable smaller learning rate by lazy random walk:[6] $U^{(k+1)} = U^{(k)} - \frac{\beta}{2} \frac{\partial \bar{\mathcal{Q}}_{smo}}{\partial U^{(k)}} = [\beta \tilde{D}^{-\frac{1}{2}} \tilde{A} \tilde{D}^{-\frac{1}{2}} + (1 - \beta) I_n] U^{(k)}$, where $\beta \in (0, 1)$ is the moving probability that a node moves to its neighbors in every period (Chung & Graham, 1997). This indicates that compared to graph convolution, lazy random walk uses a smaller learning rate. From the viewpoint of optimization, smaller learning rates slow down the convergence (Boyd et al., 2004). In the following, for the sake of uniform comparison, we rename the "lazy random walk" as "*lazy graph convolution*".

Suppose there are $K$ layers in SGC. From an optimization perspective, the back propagation procedure in SGC aims to learn a prediction function (i.e., $g()$) to minimize a softmax-based cross-entropy classification loss (denoted as $\bar{\mathcal{Q}}_{sup}$ in Eq. 3) via gradient descent, while fixing $U = (\tilde{D}^{-\frac{1}{2}} \tilde{A} \tilde{D}^{-\frac{1}{2}})^K X$. Intuitively, SGC's two parts (i.e., graph convolution and back propagation) separately reduce the Laplacian smoothing loss $\bar{\mathcal{Q}}_{smo}$ and the classification loss $\bar{\mathcal{Q}}_{sup}$ in Eq. 3. However, to consistently reduce the overall loss $\bar{\mathcal{Q}}$ in Eq. 3 via gradient descent, these two sequential parts should always guarantee to reduce the value of $\bar{\mathcal{Q}}_{sup} + \bar{\mathcal{Q}}_{smo}$. Therefore, as a whole, SGC may not guarantee to reduce the overall loss $\bar{\mathcal{Q}}$ in Eq. 3.

### 3.2 GCN ANALYSIS

The analysis of GCN is analogous to SGC. Actually, the analysis of a single-layer GCN is the same as that of SGC. Here, we consider the $k$-th layer of a multi-layer GCN.

---

[3]Here, for convenience of description, we omit the bias term.

[4]To avoid ambiguities, we use $U$ to denote the learned node embedding matrix in shallow methods, and use $U^{(k)}$ to denote the learned matrix at the $k$-th iteration.

[5]The detailed derivations can be found in Appendix D.1.

[6]The detailed derivations can be found in Appendix D.2.

Like the analysis in Section 3.1, we continue to consider a graph with self-loops, and we still adopt the symmetric normalized Laplacian matrix. Let $H^{(k)} = f(X)$ denote the learned node embedding matrix at the $k$-th layer. Then, at this layer, the objective function in Eq. 1 becomes:

$$\min_{\substack{H^{(k)}, g^{(k)} \\ H^{(0)} = X}} \hat{\mathcal{Q}}^{(k)} = \hat{\mathcal{Q}}_{sup}^{(k)} + \hat{\mathcal{Q}}_{smo}^{(k)} = \sum_{v_i \in \mathcal{V}_L} \ell(g^{(k)}(H_i^{(k)}), Y_i) + \operatorname{tr}(H^{(k)^\top} \tilde{D}^{-\frac{1}{2}} \tilde{L} \tilde{D}^{-\frac{1}{2}} H^{(k)}), \tag{4}$$

where $g^{(k)}$ stands for the mapping function at the $k$-th layer. $\hat{\mathcal{Q}}_{sup}^{(k)}$ and $\hat{\mathcal{Q}}_{smo}^{(k)}$ stand for the supervised classification loss and unsupervised Laplacian smoothing loss at the $k$-th layer, respectively.

Similarly, from an optimization perspective, at the $k$-th layer, we can rewrite the graph convolution in GCN as the standard gradient descent step: $H^{(k+1)} = H^{(k)} - \frac{1}{2} \frac{\partial \hat{\mathcal{Q}}_{smo}^{(k)}}{\partial H^{(k)}}$. On the other hand, the analysis of back propagation in GCN is more complicated than that in SGC, due to the existence of many nonlinearities and network weights. We leave this for further work. However, intuitively, at the $k$-th layer, the graph convolution in GCN only aims to reduce the Laplacian smoothing loss $\hat{\mathcal{Q}}_{smo}^{(k)}$, without considering the classification loss $\hat{\mathcal{Q}}_{sup}^{(k)}$ in Eq. 4. Nevertheless, as a prerequisite, the first graph convolution operation (the other one is back propagation) must ensure the reduction of the value of $\hat{\mathcal{Q}}_{sup}^{(k)} + \hat{\mathcal{Q}}_{smo}^{(k)}$, to consistently reduce the overall loss $\hat{\mathcal{Q}}^{(k)}$ at this layer via gradient descent. Therefore, as a whole, at the $k$-th layer, GCN may not guarantee to reduce the overall loss $\hat{\mathcal{Q}}^{(k)}$ in Eq. 4.

### 3.3 SUMMARY OF THE ANALYSIS

First of all, based on the analysis of SGC and GCN, we can get the following theorem.[7]

**Theorem 1.** *In SGC and GCN, if the node embedding results converge at the $k$-th layer, for each node pair $< v_i, v_j >$ in a connected component, we can obtain that: $U_i^{(k)} = \frac{\sqrt{D_{ii}+1}}{\sqrt{D_{jj}+1}} U_j^{(k)}$ and $H_i^{(k)} = \frac{\sqrt{D_{ii}+1}}{\sqrt{D_{jj}+1}} H_j^{(k)}$.*

Theorem 1 reveals the node embedding distribution when the over-smoothing happens in SGC and GCN. Compared to a previous similar conclusion (i.e., Theorem 2) in Oono & Suzuki (2020), our theorem gives a more precise statement. Some additional numerical verifications of Theorem 1 can be found in Appendix C.1.

In addition, the main difference of the typical GCNs compared to the classical shallow GSSL methods are summarized as follows. First, typical GCNs further consider a self-loop graph (i.e., $\tilde{A}$ in the graph convolution process defined in Eq. 2). Second, typical GCNs may involve some non-linear operations (i.e., $\sigma(\cdot)$ in the convolution process), and have a larger model capacity (if consisting of multiple trainable layers). Last, and most notably, typical GCNs may not guarantee to jointly reduce the supervised classification loss and the unsupervised Laplacian smoothing loss at each layer. In other words, typical GCNs may fail to jointly consider the graph structure and label information at each layer. This may bring new insights for designing novel GCN-type methods.

## 4 PROPOSED METHODS

In this section, we present three simple but powerful graph convolution methods in which the weight matrices and nonlinearities between consecutive layers are always removed for simplicity.

### 4.1 A SUPERVISED METHOD: OGC

The analysis in Sect. 3.1 indicates that graph convolution may only reduce the smoothing loss $\bar{\mathcal{Q}}_{smo}$, but does not reduce the supervised loss $\bar{\mathcal{Q}}_{sup}$ in Eq. 3. This motivates the proposed supervised method ***Optimized Graph Convolution* (OGC)**.

Continuing with the framework in Eq. 3, OGC alleviates this issue by jointly considering the involved two loss terms in the convolution process. Specifically, at the $k$-th iteration, it updates the learned node embeddings as follows:

$$U^{(k+1)} = U^{(k)} - \eta_{smo} \frac{\partial \bar{\mathcal{Q}}_{smo}}{\partial U^{(k)}} - \eta_{sup} \frac{\partial \bar{\mathcal{Q}}_{sup}}{\partial U^{(k)}}, \tag{5}$$

---

[7]Due to space limitation, in this paper, we give all the proofs in Appendix B.

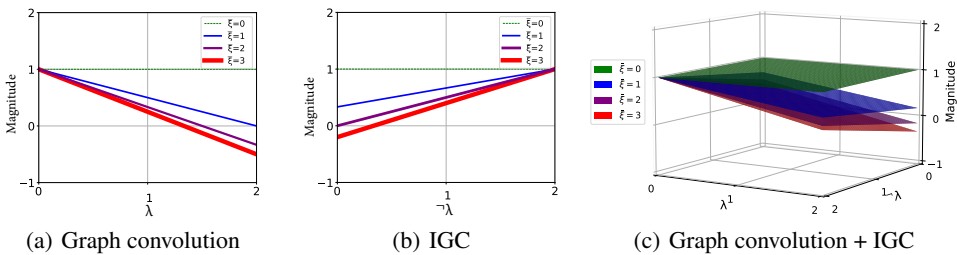

Figure 1: Filter functions. $\lambda$ and $\neg\lambda$ denote the eigenvalues of symmetric normalized Laplacian matrix of $A$ and $\neg A$, respectively.

where $\eta_{smo}$ and $\eta_{sup}$ are learning rates. For simplicity, we adopt a linear model $W \in \mathbb{R}^{d \times c}$ as the label prediction function in $\bar{\mathcal{Q}}_{sup}$. Since the learning rate $\eta_{smo}$ can be adjusted by the moving probability (i.e., $\beta$) in the lazy graph convolution, for easy understanding, we further set $\eta_{smo} = \frac{1}{2}$ and rewrite Eq. 5 in a lazy graph convolution form:

$$U^{(k+1)} = [\beta \tilde{D}^{-\frac{1}{2}} \tilde{A} \tilde{D}^{-\frac{1}{2}} + (1-\beta)I_n]U^{(k)} - \eta_{sup}S(-Y + Z^{(k)})W^\top, \tag{6}$$

where $S$ is a diagonal matrix with $S_{ii} = 1$ if node $v_i$ is labeled, and $S_{ii} = 0$ otherwise. $Z^{(k)}$ is the predicted soft labels at the $k$-th iteration, e.g., $Z^{(k)} = U^{(k)}W$ if the squared loss is used, and $Z^{(k)} = \text{softmax}(U^{(k)}W)$ if the cross-entropy loss is used. The detailed derivations can be found in Appendix D.3. In the following, for easy reference, we call the newly introduced embedding learning part (i.e., the second term in the right hand of Eq. 6) as *Supervised EmBedding (SEB)*.

**Implementation Details.** In OGC, we first initialize $U^{(0)} = X$, to satisfy the constraint in Eq. 3. Then, we alternatively update $W$ and $U$ at each iteration, as the objective function formulated in Eq. 3 leads to a multi-variable optimization problem. Specifically, when $U$ is fixed, we can update $W$ manually or by some automatic-differentiation toolboxes implemented in popular deep learning libraries. For example, if we adopt square loss for $\bar{\mathcal{Q}}_{sup}$, we can manually update $W$ as: $W = W - \eta_W \frac{\partial \bar{\mathcal{Q}}_{sup}}{\partial W} = W - \eta_W U^{(k)^\top} S(U^{(k)}W - Y)$, where $\eta_W$ is the learning rate; if we adopt an auxiliary single-layer fully-connected neural network (whose involved weight matrix is $W$ and the bias term is omitted) as the label prediction function, we can automatically update $W$ by training this auxiliary neural network. On the other hand, when $W$ is fixed, we can update $U$ via (lazy) supervised graph convolution (Eq. 6). Finally, at each iteration, we can naturally obtain the label prediction results from: $Z^{(k)} = U^{(k)}W$. For clarity, we summarize the whole procedure of OGC in Alg. 1 in Appendix.

**Time Complexity.** OGC alternatively updates $U$ and $W$ at each iteration. The first part of updating $U$ (i.e., Eq. 6) is the standard (lazy) graph convolution, with a complexity of $O(|\mathcal{E}|d)$, where $|\mathcal{E}|$ represents the number of edges in the graph. The second part (i.e., SEB) of updating $U$ costs $O(ndc)$. On the other hand, updating $W$ also costs $O(ndc)$. Supposing there are $K$ iterations, the total time complexity of OGC would be $O((|\mathcal{E}| + nc)dK)$, i.e., linear in the number of edges and nodes in the graph.

### 4.2 TWO UNSUPERVISED METHODS: GGC AND GGCM

The analysis in Sect. 3.1 shows that graph convolution in SGC only ensures the similarity between linked nodes, indicating that the dissimilarity between unlinked nodes is not considered. This motivates the proposed two unsupervised methods: *Graph Structure Preserving Graph Convolution (GGC)* and its multi-scale version *GGCM*.

#### 4.2.1 GUARANTEEING THE DISSIMILARITY BETWEEN UNLINKED NODES

We introduce a new information aggregation operator named *Inverse Graph Convolution (IGC)*, to ensure all the unlinked nodes are far away from each other in the embedding space. Because of the enormous number of unlinked node pairs, we adopt negative sampling (Mikolov et al., 2013a). Let $\neg A \in \mathbb{R}^{n \times n}$ denote a randomly generated sparse "negative" adjacency matrix in which $\neg A_{ij} = 1$ indicates that nodes $v_i$ and $v_j$ have no link, and $\neg A_{ij} = 0$ otherwise. For this objective, we maximize

the following Laplacian sharpening (Taubin, 1995) loss:

$$\max_{U, U^{(0)}=X} \mathcal{Q}_{sharp} = \sum_{i,j=1}^{n} \neg A_{ij} (\frac{1}{\sqrt{\neg D_{ii}}} U_i - \frac{1}{\sqrt{\neg D_{jj}}} U_j)^2 = \text{tr}(U^{\top} \neg D^{-\frac{1}{2}} \neg L \neg D^{-\frac{1}{2}} U), \quad (7)$$

where $\neg D$ is the degree matrix of $\neg A$, and $\neg L$ is the Laplacian matrix of $\neg A$. Here, we use a constraint: $U^{(0)} = X$, to utilize the given node attributes. This objective function can be maximized by the standard gradient ascent:

$$U^{(k+1)} = U^{(k)} + \eta_{sharp} \frac{\partial \mathcal{Q}_{sharp}}{\partial U^{(k)}} = (I_n + 2\eta_{sharp} \neg D^{-\frac{1}{2}} \neg L \neg D^{-\frac{1}{2}}) U^{(k)}, \quad (8)$$

where $\eta_{sharp}$ is the learning rate. However, as the eigenvalues of $(I_n + 2\eta_{sharp} \neg D^{-\frac{1}{2}} \neg L \neg D^{-\frac{1}{2}})$ fall in $[1, 4\eta_{sharp} + 1]$, repeated application of Eq. 8 could lead to numerical instabilities. Therefore, we modify the above equation as follows. For easy understanding, like the derivation in Eq. 6, we first rewrite Eq. 8 in a "graph convolution" form by setting the learning rate $\eta_{sharp} = \frac{1}{2}$:

$$U^{(k+1)} = (I_n + \neg D^{-\frac{1}{2}} \neg L \neg D^{-\frac{1}{2}}) U^{(k)} = (2I_n - \neg D^{-\frac{1}{2}} \neg A \neg D^{-\frac{1}{2}}) U^{(k)}. \quad (9)$$

Then, we adopt a re-normalization trick: $2I_n - \neg D^{-\frac{1}{2}} \neg A \neg D^{-\frac{1}{2}} \rightarrow \neg \tilde{D}^{-\frac{1}{2}} \neg \tilde{A} \neg \tilde{D}^{-\frac{1}{2}}$, where $\neg \tilde{A} = 2I_n - \neg A$ and $\neg \tilde{D} = 2I_n + \neg D$. As such, at the $k$-th iteration, the forward-path of IGC can be finally formulated as:

$$U^{(k+1)} = (\neg \tilde{D}^{-\frac{1}{2}} \neg \tilde{A} \neg \tilde{D}^{-\frac{1}{2}}) U^{(k)}. \quad (10)$$

Now, the spectral radius of $(\neg \tilde{D}^{-\frac{1}{2}} \neg \tilde{A} \neg \tilde{D}^{-\frac{1}{2}})$ is 1 for any choice of $\neg \tilde{A}$ (Li & Li, 2009), indicating IGC is a numerically stable operator. Actually, the newly constructed matrix $\neg \tilde{D}$ can be written as $\neg \tilde{D}_{ii} = \sum_j \text{abs}(\neg \tilde{A}_{ij})$, where $\text{abs}(\cdot)$ is the absolute value function. In other words, $\neg \tilde{D}$ acts as a normalization factor for those matrices containing both positive and negative entries.

Intuitively, compared to graph convolution, IGC optimizes a similar objective function but in an inverse direction. Likewise, its time complexity is very similar to that of graph convolution, i.e., $O(|\neg \mathcal{E}|d)$, where $|\neg \mathcal{E}|$ represents the number of edges in the randomly generated negative graph.

We now analyze the expressive power of IGC from a spectral perspective, following Balcilar et al. (2021). Theoretically, in the frequency domain, the general graph convolution operation between the signal $\mathfrak{s}$ and filter function $\Phi(\boldsymbol{\lambda})$ can be defined as: $Q \text{diag}(\Phi(\boldsymbol{\lambda}))Q^T \mathfrak{s}$, where $\text{diag}(\cdot)$ is the diagonal matrix operator, and $Q \in \mathbb{R}^{n \times d}$ and $\boldsymbol{\lambda} \in \mathbb{R}^n$ gather the eigenvectors and their corresponding eigenvalues of a Laplacian matrix, respectively.

**Theorem 2.** *The filter function of IGC can be approximated as:* $\Phi(\boldsymbol{\lambda}) \approx \frac{2-\bar{\xi}+\bar{\xi}\boldsymbol{\lambda}}{\bar{\xi}+2}$, *where $\bar{\xi}$ is the average node degree in the negative graph used in IGC.*

**Theorem 3.** *Supposing $\neg \lambda_n$ is the largest eigenvalue of matrix $\neg \tilde{D}^{-\frac{1}{2}}(\neg \tilde{D} - \neg \tilde{A})\neg \tilde{D}^{-\frac{1}{2}}$, we have that:* $\neg \lambda_n \leq 2 - \frac{4}{2+\max_i \neg D_{ii}} < 2$.

Based on Theorem 2 and Theorem 3, we can depict the filtering operation of IGC. As shown in Fig. 1(b), IGC is a high-pass filter, i.e., the response to high frequency signals will be greater than that to low frequency signals. Figure 1(b) also shows that the absolute magnitude values of IGC are always less than 1, confirming its numerically stable. In addition, as illustrated in Fig. 1(c), combining IGC and graph convolution will gather both the similarity information (among linked nodes) in the original graph and the dissimilarity information (among unlinked nodes) in the randomly generated negative graph.

### 4.2.2 GGC AND GGCM

Intuitively, by simultaneously conducting graph convolution and IGC at each iteration, we would finally preserve the original graph structure in the embedding space. Following this, we give two unsupervised methods.

**GGC: Preserving Graph Structure Information.** The idea is to preserve the graph structure during the convolution procedure. Therefore, at each iteration, GGC just simultaneously conducts (lazy) graph convolution and IGC. Specifically, before the iteration starts in GGC, we first initialize

$U^{(0)} = X$, to satisfy the constraint in the objective functions defined in Eq. 3 and Eq. 7. Then, at each (like the $k$-th) iteration, we first generate a node embedding matrix (denoted as $U_{smo}^{(k)}$) via (lazy) graph convolution. Next, we generate another node embedding matrix (denoted as $U_{sharp}^{(k)}$) via (lazy) IGC. After that, we adopt the average of these two results (i.e., $U^{(k)} = (U_{smo}^{(k)} + U_{sharp}^{(k)})/2$) as the node embedding result at this iteration. Finally, we feed the obtained $U^{(k)}$ to the next iteration.

**GGCM: Preserving Multi-scale Graph Structure Information.** The aim is to learn multi-scale representations for graph structure preserving. Inspired by DenseNet (Huang et al., 2017), we simply sum the embedding results at all iterations. Specifically, to preserve the multi-scale similarity between linked nodes, we sum the outputs of (lazy) graph convolution at each iteration. On the other hand, to preserve the multi-scale dissimilarity between unlinked nodes, we also sum the outputs of (lazy) IGC at each iteration. In addition, to be consistent with the multi-scale information of the first part, we use the same inputs for IGC as in the first part. As a whole, at the $k$-th iteration, GGCM gathers the multi-scale representations (denoted as $U_M{}^{(k)}$) of graph structure information, as follows: $U_M{}^{(k)} = \alpha X + (1-\alpha)\frac{1}{k}\sum_{t=1}^{k}[(U_{smo}^{(t)} + U_{sharp}^{(t)})/2]$, where $\alpha$ is a balancing hyper-parameter.

We can see that these two methods are "no-learning" methods, as they do not need to learn any parameters. For clarity, we summarize the whole procedure of GGC and GGCM method in Alg. 2 and Alg. 3 in Appendix, respectively. Some rigorous analysis can be found in Appendix B.4.

**Time Complexity.** At each iteration, GGC and GGCM both generate two parts of node embedding results. The first part is obtained via the standard graph convolution whose time complexity is $O(|\mathcal{E}|d)$. The other part is obtained via IGC whose time complexity is $O(|\neg\mathcal{E}|d)$. Supposing there are $K$ iterations, the overall time complexity of these two methods are both $O((|\mathcal{E}| + |\neg\mathcal{E}|)dK)$, i.e., linear in the number of graph and negative graph edges.

## 5 EXPERIMENTS

### 5.1 EXPERIMENTAL SETUP

**Datasets.** For semi-supervised node classification, we first adopt three well-known citation network datasets: Cora, Citeseer and Pubmed (Sen et al., 2008), with the standard train/val/test split setting (Kipf & Welling, 2017; Yang et al., 2016). Experiments on these citation networks are transductive, i.e., all nodes are accessible during training. For inductive learning, we use a much larger social graph Reddit (Hamilton et al., 2017). In this graph, testing nodes are unseen during the model learning process.

**Baselines.** We compare our methods with some unsupervised methods including the skip-gram based methods (DeepWalk (Perozzi et al., 2014) and node2vec (Grover & Leskovec, 2016)), autoencoder based methods (GAE (Kipf & Welling, 2016) and VERSE (Tsitsulin et al., 2018)), graph contrastive methods (DGI (Velickovic et al., 2019), BGRL (Thakoor et al., 2022), and N2N (Dong et al., 2022) ) and some GCN methods (SGC (Wu et al., 2019), S²GC (Zhu & Koniusz, 2021) and G²CN (Li et al., 2022)). In addition, we further compare some supervised methods which can be mainly categorized into three types. The first type are shallow cluster-assumption based GSSL methods: label propagation (LP) (Zhu et al., 2003) and manifold regularization (ManiReg) (Belkin et al., 2006). The second type are some shallow GCN methods, including GCN (Kipf & Welling, 2017), APPNP (Klicpera et al., 2019), Eigen-GCN (Zhang et al., 2021b), GNN-LF/HF (Zhu et al., 2021), C&S (Huang et al., 2021), NDLS (Zhang et al., 2021a), ChebNetII (He et al., 2022) and OAGS (Song et al., 2022b). Finally, we also compare with some deep GCN methods: JKNet (Xu et al., 2018b), Incep (Rong et al., 2019), GCNII (Chen et al., 2020), GRAND (Feng et al., 2020) and ACMP (Wang et al., 2023).

**Implementation Details.** In our methods, we always set the maximum iteration number as 64. As OGC is actually a shallow method, it does not need the validation set to avoid overfitting. Therefore, like LP (Zhu et al., 2003) and C&S (Huang et al., 2021), we use both train and validation sets as the supervised knowledge. In addition, to further mitigate the risk of overfitting within the learned node embeddings, when updating $U$ in Eq. 6, we only use the train set to build the label indicator diagonal matrix $S$. We call this trick "*Less Is More (LIM)*". In our two unsupervised methods GGC and GGCM, after obtaining the node embeddings, we train a linear logistic regression classifier for label prediction on the training set, and conduct a grid search to tune all hyper-parameters on the validation set. More experimental details can be found in Appendix A.

|  | Method | #Layer | Cora | Citeseer | Pubmed |
|---|---|---|---|---|---|
| **supervised** | LP | – | 71.5 | 48.9 | 65.8 |
|  | ManiReg | – | 59.5 | 60.1 | 70.7 |
|  | GCN | 2 | 81.5 | 70.3 | 79.0 |
|  | APPNP | 2 | 83.3 | 71.8 | 80.1 |
|  | Eigen-GCN | 2 | 78.9± 0.7 | 66.5±0.3 | 78.6±0.1 |
|  | GNN-LF/HF | 2 | 84.0±0.2 | 72.3±0.3 | 80.5±0.3 |
|  | C&S | 3 | 84.6±0.5 | 75.0±0.3 | 81.2±0.4 |
|  | NDLS | 2 | 84.6±0.5 | 73.7±0.6 | 81.4±0.4 |
|  | ChebNetII | 2 | 83.7±0.3 | 72.8±0.2 | 80.5±0.2 |
|  | OAGS | 2 | 83.9±0.5 | 73.7±0.7 | 81.9±0.9 |
|  | JKNet | {4, 16, 32} | 82.7 ± 0.4 | 73.0 ± 0.5 | 77.9 ± 0.4 |
|  | Incep | {64, 4, 4} | 82.8 | 72.3 | 79.5 |
|  | GCNII | {64, 32, 16} | 85.5±0.5 | 73.4 ± 0.6 | 80.2 ± 0.4 |
|  | GRAND | {8, 2, 5} | 85.4±0.4 | 75.4±0.4 | 82.7±0.6 |
|  | ACMP | {8, 4, 32} | 84.9±0.6 | 75.5 ± 1.0 | 79.4±0.4 |
|  | **OGC** (ours) | > 2 | **86.9 ± 0.0** | **77.5 ± 0.0** | **83.4 ± 0.0** |
| **unsupervised** | DeepWalk | – | 67.2 | 43.2 | 65.3 |
|  | node2vec | – | 71.5 | 45.8 | 71.3 |
|  | VERSE | – | 72.5 ± 0.3 | 55.5 ± 0.4 | – |
|  | GAE | 4 | 71.5 ± 0.4 | 65.8 ± 0.4 | 72.1 ± 0.5 |
|  | DGI | 2 | 82.3 ± 0.6 | 71.8 ± 0.7 | 76.8 ± 0.6 |
|  | DGI Random-Init | 2 | 69.3 ± 1.4 | 61.9 ± 1.6 | 69.6 ± 1.9 |
|  | BGRL | 2 | 81.1±0.2 | 71.6±0.4 | 80.0±0.4 |
|  | N2N | 2 | 83.1±0.4 | 73.1±0.6 | 80.1±0.7 |
|  | SGC | 2 | 81.0 ±0.0 | 71.9 ±0.1 | 78.9 ±0.0 |
|  | S²GC | 16 | 83.0 ±0.2 | 73.6 ±0.1 | 80.2 ±0.2 |
|  | G²CN | 10 | 82.7 | 73.8 | 80.4 |
|  | **GGC** (ours) | > 2 | 81.8 ± 0.3 | 73.8 ± 0.2 | 79.6 ± 0.1 |
|  | **GGCM** (ours) | > 2 | **83.6 ± 0.2** | **74.2 ± 0.1** | **80.8 ± 0.1** |

Table 1: Classification accuracy (%). The "#Layers" column highlights the best layer number.

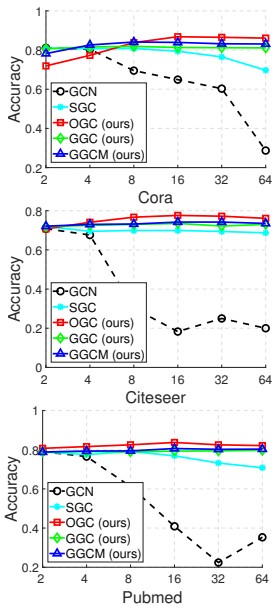

Figure 2: Classification performance w.r.t. layer number ($x$-axis: layer number).

## 5.2 TRANSDUCTIVE NODE CLASSIFICATION

Table 1 summarizes the classification results. Firstly, we can clearly see that our supervised method OGC consistently outperforms all baselines by a large margin. This indicates that the ability to include validation labels is an advantage of our approach. Secondly, our two unsupervised methods GGC and GGCM are also very powerful; especially, GGCM achieves very competitive performance on all three datasets. It's worth noting that very deep models, such as GCNII and GRAND, typically involve huge amounts of parameters to learn, contain lots of hyper-parameters to tune, and take a long time to train. Moreover, they all have to retrain the model totally, once the layer number is fixed. On the contrary, our three methods, which all run in an iterative way with very few parameters, do not have these issues. Finally, we also highlight that: GGC and GGCM greatly outperform Eigen-GCN whose experimental results show that preserving graph structure information in GCN-type models may hurt the performance. In contrast, our work evidently demonstrates this value.

**The Effect of Depths/Iterations.** As shown in Fig. 2, in most cases, the performance of our methods increases as the iteration goes on. Specifically, they all tend to get the best performance at around 16-th or 32-th iteration, and could maintain similar performance as the iteration number increases to 64. In addition, we give the detailed results of depth/iteration effect experiment in Appendix C.2, an efficiency test in Appendix C.3, supervised label setting test in Appendix C.4, and various setting test of our methods in Appendix C.5.

## 5.3 INDUCTIVE NODE CLASSIFICATION

For baselines, besides the above-mentioned methods GCN, DGI, SGC and S2GC, we further adopt GaAN (Zhang et al., 2018), NDLS (Zhang et al., 2021a), supervised and unsupervised variants of GraphSAGE (Hamilton et al., 2017) and FastGCN (Chen et al., 2018). We also reuse the metrics already reported in Wu et al. (2019) and Chen et al. (2018). Table 2 shows the averaged test results. As expected, our supervised method OGC could still get the best performance. In addition, our unsupervised methods GGC and GGCM still outperform SGC, S2GC, all other unsupervised methods, and even some supervised GNNs.

## 5.4 GRAPH RECONSTRUCTION

Following Tsitsulin et al. (2018), to avoid interference, we only use graph structure information by setting $X = I_n$. Then, for each node, we rank all other nodes according to their distance to this one

| Setting | Method | Test F1 |
|---------|--------|---------|
| **Supervised** | GaAN
SAGE-mean
SAGE-LSTM
SAGE-GCN
FastGCN
GCN
NDLS
**OGC** (ours) | 96.4
95.0
95.4
93.0
93.7
OOM
96.8
**97.9** |
| **Unsupervised** | SAGE-mean
SAGE-LSTM
SAGE-GCN
DGI
SGC
S$^2$GC
**GGC** (ours)
**GGCM** (ours) | 89.7
90.7
90.8
94.0
94.9
95.3
95.0
**95.8** |

Table 2: Micro-averaged F1 (%) on Reddit.

| Dataset | Method | Layers/Iterations | | | | | |
|---------|--------|------|------|------|------|------|------|
| | | 2 | 4 | 8 | 16 | 32 | 64 |
| Cora | SGC
S$^2$GC
GGC
GGCM | 68.15
84.84
81.19
**88.97** | 60.47
80.49
81.22
86.92 | 50.38
76.74
81.22
84.98 | 41.76
73.87
81.22
83.93 | 31.93
72.55
81.22
82.96 | 20.51
72.05
81.22
82.94 |
| Citeseer | SGC
S$^2$GC
GGC
GGCM | 74.89
90.66
88.66
**91.94** | 58.26
88.67
88.87
91.62 | 50.84
84.99
88.90
91.11 | 44.56
78.04
88.90
90.64 | 37.60
71.10
88.90
90.10 | 31.16
68.17
88.90
89.98 |
| Pubmed | SGC
S$^2$GC
GGC
GGCM | 39.57
56.00
46.88
56.95 | 27.81
54.63
49.48
**57.90** | 17.12
50.65
49.96
57.63 | 13.06
48.78
50.00
56.82 | 8.72
48.14
50.00
56.47 | 4.341
47.07
50.00
55.90 |

Table 3: Graph reconstruction accuracy.

in the embedding space based on Euclidean distance. Then, we take the same number of nodes equal to its actual degree as its predicted neighbors. Next, we count the rate of correct predictions. Finally, the graph reconstruction accuracy is computed as the average over all nodes. For comparison, we adopt two no-learning baselines: SGC and S$^2$GC. As shown in Table 3, the performance of SGC and S$^2$GC declines quickly as the iteration number increases. In contrast, our two methods GGC and GGCM perform much better. Especially, by gathering the multi-scale information, GGCM always obtains the best performance.

## 6 RELATED WORK

In this section, we review some related work on understanding GNNs and fixing the over-smoothing problem. More comparison and related work discussion can be found in Appendix E.

**Understanding GNNs.** Generally, existing studies can be categorized into two groups. The first group analyses the intrinsic properties of GNNs, including the equivalence of GNNs to the Weisfeiler-Lehman test (Xu et al., 2018a), the low-pass filtering feature of GNNs (NT & Maehara, 2019), and the expressive power of K-hop message passing in GNNs (Feng et al., 2023). The second group connects GNNs to non GNN-type graph learning methods, such as traditional graph kernels (Fu et al., 2020; Zhu et al., 2021; Ma et al., 2021), classical iterative algorithms (Yang et al., 2021), and implicit layers and nonlinear diffusion steps (Chen et al., 2022). This study falls under the second group. However, unlike existing discussions limited to the graph convolution process of GNNs, we additionally consider the label learning part, by integrating these two parts into a unified optimization framework.

**Fixing the Over-smoothing Problem.** Recent studies on this problem mainly follow two lines. The first line is to make deep GNNs trainable. For example, JKNet (Xu et al., 2018b) uses skip-connection, DropEdge (Rong et al., 2019) suggests randomly removing out some graph edges, GC-NII (Chen et al., 2020) adopts initial residual and identity mapping, and OrderedGNN (Song et al., 2023) adopts ordering message passing. On the other hand, some approaches try to combine deep propagation with shallow neural networks. For example, SGC (Wu et al., 2019) and S$^2$GC (Zhu & Koniusz, 2021) use the $K$-th power of the graph convolution matrix, and APPNP (Klicpera et al., 2019) adopts personalized PageRank (Page et al., 1999). In comparison, our work is inspired by the fact that traditional GSSL methods do not have the over-smoothing problem, and tries to fix this problem within a unified GSSL optimization framework.

## 7 CONCLUSION

In this work, by using a unified GSSL optimization framework, we find that compared to traditional GSSL methods, the learning process of typical GCNs may fail to jointly consider the graph structure and label information at each layer. Motivated by this, we introduce two novel operators (SEB and IGC), based on which we further propose three simple but powerful graph convolution methods. Extensive experiments demonstrate that our methods exhibit superior performance against state-of-the-art methods. We believe that introducing supervised information or preserving the graph structure in the convolution process of deep GCNs can bring substantial improvement, especially considering the computational overhead that most GCNs require.

## REPRODUCIBILITY STATEMENT

To ensure the reproducibility of the empirical results, we include our main code in the supplementary material. All the experimental details are given in Sect. 5.1 and Appendix A. The complete proof of all theoretical results presented in the main paper can be found in Appendix B. The detailed derivations of some important equations are also given in Appendix D.

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

# Appendices

## A  DATASETS AND IMPLEMENTATION DETAILS

Table 4: Summary of the datasets.

| Dataset | Task | Nodes | Edges | Features | Classes | Train/Val/Test Nodes |
|---------|------|-------|-------|----------|---------|----------------------|
| **Cora** | Transductive | 2,708 | 5,429 | 1,433 | 7 | 140/500/1,000 |
| **Citeseer** | Transductive | 3,327 | 4,732 | 3,703 | 6 | 120/500/1,000 |
| **Pubmed** | Transductive | 19,717 | 44,338 | 500 | 3 | 60/500/1,000 |
| **Reddit** | Inductive | 232,965 | 11,606,919 | 602 | 41 | 152,410/23,699/55,334 |

### A.1  DATASETS

Table 4 summarizes the statistics of the used three citation networks (Cora, Citeseer, and Pubmed (Sen et al., 2008)) and one social network Reddit (Hamilton et al., 2017). In these three citation networks, nodes are documents, edges are citations, and each node feature is the bag-of-words representation of the document belonging to one of the topics. We adopt the standard data-split setting and feature preprocessing used in Kipf & Welling (2017) and Yang et al. (2016). Experiments on these citation networks are transductive, i.e., all nodes are accessible during training. For inductive learning, we use a much larger social network dataset Reddit (Hamilton et al., 2017). We also adopt the standard data-split setting and feature preprocessing used in Wu et al. (2019). In this graph, testing nodes are unseen during the model learning process.

### A.2  IMPLEMENTATION DETAILS

In all our methods, we always set the maximum iteration number to 64. Specifically, in our supervised method OGC, we adopt squared loss in its convolution process (defined in Eq. 5). As OGC is actually a shallow method, it does not need the validation set to avoid overfitting. Therefore, like LP (Zhu et al., 2003) and C&S (Huang et al., 2021), we use both training and validation sets as the supervised knowledge. In addition, to further avoid the overfitting rick of the learned node embedding, when updating $U$ in Eq. 6, we only use the train set to build the label indicator diagonal matrix $S$. We call this trick "***Less Is More (LIM)***". All its hyper-parameters are chosen to consistently improve the classification accuracy on the labeled set during the iteration process. At last, we will stop its iteration when the predicted labels converge.

In our two unsupervised methods GGC and GGCM, after obtaining the node embeddings at each iteration, we train a classifier for label prediction. Specifically, like Wu et al. (2019) and Zhu & Koniusz (2021), we train a linear logistic regression classifier for 100 epochs, with weight decay, Adam (Kingma & Ba, 2014) and learning rate 0.2. Besides, we also iteratively adjust the moving probability values in these two methods by multiplying $\beta$ with a decline rate. In these two methods, we both conduct a grid search to tune all hyper-parameters on the validation set. Especially, for the used negative graph $\neg A$ in IGC, the edge number is tuned among $\{|\mathcal{E}|, 10|\mathcal{E}|, 20|\mathcal{E}|, 30|\mathcal{E}|, 40|\mathcal{E}|, 50|\mathcal{E}|\}$, where $|\mathcal{E}|$ is the edge number of the original graph. At last, the node embeddings together with the corresponding classifier, which obtain the best performance on the validation set, are used in the subsequent experiments.

Our codes are all written in Python 3.8.3, PyTorch 1.7.1, NumPy 1.16.4, SciPy 1.3.0, and CUDA 11.0. All experiments are conducted for 100 trials with random seeds.

## B  THEORETICAL ANALYSIS

### B.1 PROOF OF THEOREM 1

*Proof.* In SGC, as conducting graph convolution only affects the Laplacian smoothing loss (i.e., $\bar{\mathcal{Q}}_{smo}$) in Eq. 3, in the following, we only consider this loss term. In a connected component, setting $U_i^{(k)} = \sqrt{D_{ii}+1}\overrightarrow{\tau} = \sqrt{\tilde{D}_{ii}}\overrightarrow{\tau}$ and $U_j^{(k)} = \sqrt{D_{jj}+1}\overrightarrow{\tau} = \sqrt{\tilde{D}_{jj}}\overrightarrow{\tau}$ will lead the involved Laplacian smoothing loss part in this component to reach 0, where $\overrightarrow{\tau}$ is an arbitrary $d$-dimensional vector. Repeating this setting in every connected graph component will lead the loss $\bar{\mathcal{Q}}_{smo}$ in Eq. 3 to reach its lower bound (i.e., 0), which proves the theorem in SGC.

In GCN, the analysis is analogous to that in SGC. Specifically, we can similarly prove this theorem by setting $H_i^{(k)} = \sqrt{D_{ii}+1}\overrightarrow{\tau^{(k)}} = \sqrt{\tilde{D}_{ii}}\overrightarrow{\tau^{(k)}}$ and $H_j^{(k)} = \sqrt{D_{jj}+1}\overrightarrow{\tau^{(k)}} = \sqrt{\tilde{D}_{jj}}\overrightarrow{\tau^{(k)}}$ to reach the lower bound (i.e., 0) of $\hat{\mathcal{Q}}_{smo}^{(k)}$ defined in Eq. 4, where $\overrightarrow{\tau^{(k)}}$ is an arbitrary $d^{(k)}$-dimensional vector at the $k$-th iteration. $\qquad\square$

### B.2 PROOF OF THEOREM 2

*Proof.* First of all, we consider a "negative" regular graph whose degree is $\xi$. We can write its diagonal degree matrix as $\neg D = \xi I_n$, and write its symmetrically normalized Laplacian matrix as $\neg L_{sym} = I_n - \neg A/\xi$ (and $\neg A = \xi I_n - \xi \neg L_{sym}$). According to the definition of IGC (in Eq. 10), we can get:

$$
\begin{aligned}
\neg\tilde{D}^{-\frac{1}{2}}\neg\tilde{A}\neg\tilde{D}^{-\frac{1}{2}} &= \neg\tilde{D}^{-\frac{1}{2}}(2I_n - \neg A)\neg\tilde{D}^{-\frac{1}{2}} \\
&= \frac{2I_n - \xi I_n + \xi\neg L_{sym}}{\xi+2} \quad \text{(By replacing variables)} \\
&= Q\left(\frac{(2-\xi)I_n + \xi\,\text{diag}(\boldsymbol{\lambda})}{\xi+2}\right)Q^T.
\end{aligned}
$$ 
(11)

Therefore, w.r.t. this regular graph, we can get the following filter function:

$$
\Phi(\boldsymbol{\lambda}) = \frac{2-\xi+\xi\boldsymbol{\lambda}}{\xi+2}.
$$ 
(12)

Assuming that the node degree distribution is uniform, we can get the approximation: $\xi \approx \bar{\xi} = \frac{1}{n}\sum_{i=1}^{n}\neg D_{ii}$. Likewise, we can obtain an approximation of its filter function as a function of the average node degree by replacing $\xi$ with $\bar{\xi}$ in the above equation, which proves the theorem.

$\qquad\square$

### B.3 PROOF OF THEOREM 3

*Proof.* Let $\epsilon_1 \le \epsilon_2 \le \cdots \le \epsilon_n$ denote the eigenvalues of $\neg D^{-\frac{1}{2}}\neg A\neg D^{-\frac{1}{2}}$ and $\omega_1 \le \omega_2 \le \cdots \le \omega_n$ denote the eigenvalues of $\neg\tilde{D}^{-\frac{1}{2}}\neg A\neg\tilde{D}^{-\frac{1}{2}}$. It is easy to know $\epsilon_n = 1$. In addition, by choosing $x$ such that $||x|| = 1$ and $y = \neg D^{\frac{1}{2}}\neg\tilde{D}^{-\frac{1}{2}}x$, we can get $||y||^2 = \sum_i \frac{\neg D_{ii}}{\neg D_{ii}+2}x_i^2$ and $\frac{\min_i \neg D_{ii}}{2+\min_i \neg D_{ii}} \le ||y||^2 \le \frac{\max_i \neg D_{ii}}{2+\max_i \neg D_{ii}}$.

First of all, we use the Rayleigh quotient to provide a higher bound to $\omega_n$:

$$
\begin{aligned}
\omega_n &= \max_{||x||=1} \left( x^T \neg \tilde{D}^{-\frac{1}{2}} \neg A \neg \tilde{D}^{-\frac{1}{2}} x \right) \\
&= \max_{||x||=1} \left( y^T \neg D^{-\frac{1}{2}} \neg A \neg D^{-\frac{1}{2}} y \right) \quad \text{(By replacing variable)} \\
&= \max_{||x||=1} \left( \frac{y^T \neg D^{-\frac{1}{2}} \neg A \neg D^{-\frac{1}{2}} y}{||y||^2} ||y||^2 \right) \\
&\because \max(\mathfrak{A}\mathfrak{B}) = \max(\mathfrak{A})\max(\mathfrak{B}) \text{ if } \max(\mathfrak{A}) > 0, \forall \mathfrak{B} > 0; \\
&\text{and } \max_{||x||=1} \left( y^T \neg D^{-\frac{1}{2}} \neg A \neg D^{-\frac{1}{2}} y / ||y||^2 \right) = \epsilon_n > 0: \\
&= \epsilon_n \max_{||x||=1} ||y||^2 \\
&\leq \frac{\max_i \neg D_{ii}}{2 + \max_i \neg D_{ii}} \quad \text{(See the properties of } ||y||^2\text{)}
\end{aligned}
\tag{13}
$$

Then, by rewriting matrix $\neg \tilde{D}^{-\frac{1}{2}}(\neg \tilde{D} - \neg \tilde{A})\neg \tilde{D}^{-\frac{1}{2}}$ as $I_n - \neg \tilde{D}^{-\frac{1}{2}}(2I_n - \neg A)\neg \tilde{D}^{-\frac{1}{2}}$, we can get:

$$
\begin{aligned}
\neg \lambda_n &= \max_{||x||=1} x^T \left( I_n - 2\neg \tilde{D}^{-1} + \neg \tilde{D}^{-\frac{1}{2}} \neg A \neg \tilde{D}^{-\frac{1}{2}} \right) x \\
&\leq 1 - \min_{||x||=1} 2x^T \neg \tilde{D}^{-1} x + \max_{||x||=1} \left( x^T \neg \tilde{D}^{-\frac{1}{2}} \neg A \neg \tilde{D}^{-\frac{1}{2}} x \right) \\
&\because \omega_n \text{ is the largest eigenvalue of } \neg \tilde{D}^{-\frac{1}{2}} \neg A \tilde{D}^{-\frac{1}{2}} \\
&= 1 - \frac{2}{2 + \max_i \neg D_{ii}} + \omega_n \\
&\leq 1 - \frac{2}{2 + \max_i \neg D_{ii}} + \frac{\max_i \neg D_{ii}}{2 + \max_i \neg D_{ii}} \\
&\text{(The higher bound of } \omega_n \text{ shown in Eq. 13)} \\
&= 2 - \frac{4}{2 + \max_i \neg D_{ii}}.
\end{aligned}
\tag{14}
$$

$\square$

## B.4 THEORETICAL ANALYSIS OF GGC

**Theorem 4.** *IGC (defined in Eq. 10) can be reformulated as a gradient descent procedure:* $U^{(k+1)} = U^{(k)} - \frac{1}{2}\frac{\partial \mathcal{Q}_{igc}}{\partial U^{(k)}}$, *where* $\mathcal{Q}_{igc} = \text{tr}(U^\top \neg \tilde{D}^{-\frac{1}{2}}(\neg \tilde{D} - \neg \tilde{A})\neg \tilde{D}^{-\frac{1}{2}}U)$.

*Proof.* We can minimize the loss $\mathcal{Q}_{igc}$ in Theorem 4 by the standard gradient descent:

$$
\begin{aligned}
U^{(k+1)} &= U^{(k)} - \eta_{igc}\frac{\partial \mathcal{Q}_{igc}}{\partial U^{(k)}} \\
&= U^{(k)} - 2\eta_{igc}\neg \tilde{D}^{-\frac{1}{2}}(\neg \tilde{D} - \neg \tilde{A})\neg \tilde{D}^{-\frac{1}{2}}U^{(k)},
\end{aligned}
\tag{15}
$$

where $\eta_{igc}$ is the learning rate in this part. If we set $\eta_{igc} = \frac{1}{2}$, we can get:

$$
\begin{aligned}
U^{(k+1)} &= U^{(k)} - \neg \tilde{D}^{-\frac{1}{2}}(\neg \tilde{D} - \neg \tilde{A})\neg \tilde{D}^{-\frac{1}{2}}U^{(k)} \\
&= (\neg \tilde{D}^{-\frac{1}{2}} \neg \tilde{A} \neg \tilde{D}^{-\frac{1}{2}})U^{(k)}.
\end{aligned}
\tag{16}
$$

The above equation is exactly the convolution equation (i.e., Eq. 10) used in IGC. Therefore, the theorem is proved. $\square$

**Theorem 5.** *The objective function of GGC is:* $\min_{U, U^{(0)}=X} \mathcal{Q}_{ggc} = (\bar{\mathcal{Q}}_{smo} + \mathcal{Q}_{igc})/2$, *where* $\bar{\mathcal{Q}}_{smo}$ *is the Laplacian smoothing loss defined in Eq. 3 and* $\mathcal{Q}_{igc}$ *is the loss function of IGC defined in Theorem 4.*

*Proof.* We can minimize the loss $\mathcal{Q}_{ggc}$ in Theorem 5 by the standard gradient descent:

$$
\begin{aligned}
U^{(k+1)} &= U^{(k)} - \frac{\eta_{smo}}{2}\frac{\partial \bar{\mathcal{Q}}_{smo}}{\partial U^{(k)}} - \frac{\eta_{igc}}{2}\frac{\partial \mathcal{Q}_{igc}}{\partial U^{(k)}} \\
&= \frac{U^{(k)}}{2} - \frac{\eta_{smo}}{2}\frac{\partial \bar{\mathcal{Q}}_{smo}}{\partial U^{(k)}} + \frac{U^{(k)}}{2} - \frac{\eta_{igc}}{2}\frac{\partial \mathcal{Q}_{igc}}{\partial U^{(k)}} \\
&= \frac{U^{(k)} - \eta_{smo}\frac{\partial \bar{\mathcal{Q}}_{smo}}{\partial U^{(k)}}}{2} + \frac{U^{(k)} - \eta_{igc}\frac{\partial \bar{\mathcal{Q}}_{igc}}{\partial U^{(k)}}}{2},
\end{aligned}
\tag{17}
$$

where $\eta_{smo}$ and $\eta_{igc}$ are the learning rates.

Those two parts in the last line of Eq. 17 are exactly the one half of the results of (lazy) graph convolution and IGC. Therefore, this theorem is proved. $\qquad\square$

---

**Algorithm 1** OGC

---

**Require:** Graph information ($A$ and $X$), the max iteration number $K$, and the label information of a small node set $\mathcal{V}_L$

**Ensure:** The label predictions

1: Initialize $U^{(0)} = X$ and $k = 0$
2: **repeat**
3:     $k = k + 1$
4:     Update $W$ manually or by some automatic-differentiation toolboxes (Sect. 4.1)
5:     Update $U^{(k)}$ via (lazy) supervised graph convolution (Eq. 6)
6:     Get the label predictions $\hat{Y}^{(k)}$ from: $Z^{(k)} = U^{(k)}W$
7: **until** $\hat{Y}^{(k)}$ converges or $k \geq K$
8: **return** $\hat{Y}^{(k)}$

---

**Algorithm 2** GGC

---

**Require:** Graph information ($A$ and $X$), the max iteration number $K$, and moving out probability $\beta$

**Ensure:** The learned node embedding result set $\{U^{(k)}|k = 0 : K\}$

1: Initialize $U^{(0)} = X$
2: **for** $k = 1$ to $K$ **do**
3:     Get $U_{smo}^{(k)}$ via (lazy) graph convolution
4:     Get $U_{sharp}^{(k)}$ via (lazy) IGC (Eq. 10)
5:     Get $U^{(k)} = (U_{smo}^{(k)} + U_{sharp}^{(k)})/2$
6:     Decline $\beta$ with a decay factor
7: **end for**
8: **return** $\{U^{(0)}, U^{(1)}, ..., U^{(K)}\}$

---

**Algorithm 3** GGCM

---

**Require:** Graph information ($A$ and $X$), the max iteration number $K$, and moving out probability $\beta$

**Ensure:** The learned multi-scale node embedding result set $\{U_M{}^{(k)}|k = 0 : K\}$

1: Initialize $U^{(0)} = X$
2: **for** $k = 1$ to $K$ **do**
3:     Get $U_{smo}^{(k)}$ via (lazy) graph convolution
4:     Get $U_{sharp}^{(k)}$ via (lazy) IGC (Eq. 10)
5:     Set $U^{(k)} = U_{smo}^{(k)}$
6:     $U_M{}^{(k)} = \alpha X + (1 - \alpha)\frac{1}{k}\sum_{t=1}^{k}[(U_{smo}^{(t)} + U_{sharp}^{(t)})/2]$
7:     Decline $\beta$ with a decay factor
8: **end for**
9: **return** $\{U_M{}^{(0)}, U_M{}^{(1)}, ..., U_M{}^{(K)}\}$

---

# C    ADDITIONAL EXPERIMENTS

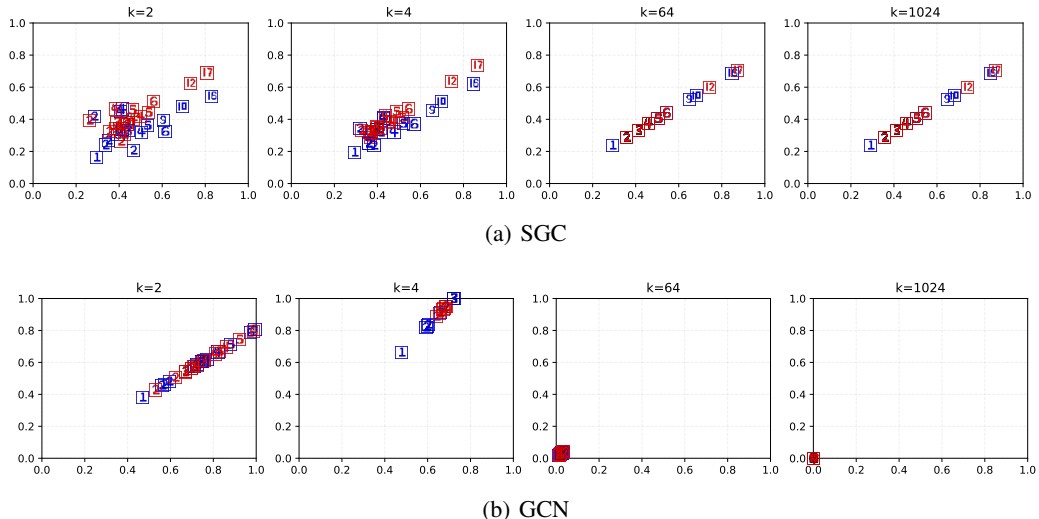

Figure 3: Embedding visualization on Zachary's karate club network. Colors denote class labels, and numbers (in squares) denote node degrees.

Table 5: Parts of the detailed node embedding results in Zachary's karate club network.

| No. (Deg.) | SGC (k=64/1024) | GCN (k=64) | GCN (k=1024) |
|---|---|---|---|
| 11 (1) | $[.2904, .2344]$ | $[.0111, .0137]$ | $[1.09e-20, 4.39e-21]$ |
| ... | ... | ... | ... |
| 4 (3) | $[.4107, .3314]$ | $[.0157, .0193]$ | $[1.53e-20, 6.21e-21]$ |
| 10 (3) | $[.4107, .3314]$ | $[.0157, .0193]$ | $[1.53e-20, 6.21e-21]$ |
| ... | ... | ... | ... |
| 5 (4) | $[.4591, .3705]$ | $[.0175, .0216]$ | $[1.72e-20, 6.94e-21]$ |
| 6 (4) | $[.4591, .3705]$ | $[.0175, .0216]$ | $[1.72e-20, 6.94e-21]$ |
| 7 (4) | $[.4591, .3705]$ | $[.0175, .0216]$ | $[1.72e-20, 6.94e-21]$ |
| ... | ... | ... | ... |
| 0 (16) | $[.8466, .6832]$ | $[.0323, .0399]$ | $[3.16e-20, 1.28e-20]$ |
| 33 (17) | $[.8712, .7031]$ | $[.0333, .0410]$ | $[3.26e-20, 1.32e-20]$ |

## C.1    NUMERICAL VERIFICATION OF THEOREM 1

To verify our theoretical analysis, we conduct a numerical verification experiment on Zachary's karate club network. This graph has two classes (groups) with 34 nodes connected by 154 (undirected and unweighted) edges. As this graph has no node attributes, we randomly generate two dimensional features for each node. Then, we test SGC and GCN by varying the number of layers in these methods. Specifically, for GCN, we set the dimensions of all hidden layers to two, use ReLU activation, and adopt uniform weight initialization. At last, in these two methods, we always use the outputs of the last layer as the final node embedding results.

Figure 3 shows the visualization of the obtained node embeddings. For a better verification, we also list parts of the corresponding numerical results in Table 5. Here, we combine the results of SGC with $k = 64$ and $k = 1024$ to one column, since their results are exactly the same. We can clearly find that in both SGC and GCN, the nodes (in the same connected component) with the same degree tend to converge to the same embedding results. We also note that the convergence in GCN is less obvious than that in SGC, which may be due to the existence of many nonlinearities and network weights in GCN. In addition, we can see that the ratio of the embeddings of nodes $v_i$

Table 6: Classification results (%) w.r.t. various layer/iteration numbers. **OOM**: Out of memory.

| Dataset | Method | Layers/Iterations | | | | | |
|---|---|---|---|---|---|---|---|
| | | 2 | 4 | 8 | 16 | 32 | 64 |
| **Cora** | GNN-LF/HF | 81.3 | 83.0 | 83.5 | 83.9 | 83.8 | 83.7 |
| | GCN | 81.1 | 80.4 | 69.5 | 64.9 | 60.3 | 28.7 |
| | GCN(Drop) | 82.8 | 82.0 | 75.8 | 75.7 | 62.5 | 49.5 |
| | JKNet | - | 80.2 | 80.7 | 80.2 | 81.1 | 71.5 |
| | JKNet(Drop) | - | 83.3 | 82.6 | 83.0 | 82.5 | 83.2 |
| | Incep | - | 77.6 | 76.5 | 81.7 | 81.7 | 80.0 |
| | Incep(Drop) | - | 82.9 | 82.5 | 83.1 | 83.1 | 83.5 |
| | GCNII | 82.2 | 82.6 | 84.2 | 84.6 | 85.4 | 85.5 |
| | GRAND | 83.8 | 84.5 | 85.4 | 84.0 | 82.9 | 79.6 |
| | ACMP | 82.0 | 83.9 | 84.0 | 83.2 | 83.1 | 80.5 |
| | **OGC** (ours) | 71.9 | 77.3 | 83.8 | **87.0** | 86.8 | 86.5 |
| | SGC | 81.0 | 80.7 | 80.8 | 79.4 | 76.5 | 69.7 |
| | S$^2$GC | 78.8 | 81.4 | 82.2 | 82.5 | 82.3 | 81.2 |
| | **GGC** (ours) | 80.6 | 81.6 | 81.9 | 81.3 | 81.2 | 81.1 |
| | **GGCM** (ours) | 78.2 | 80.9 | **84.1** | 83.9 | 83.5 | 83.1 |
| **Citeseer** | GNN-LF/HF | 71.1 | 71.5 | 72.0 | 72.1 | 72.2 | 72.2 |
| | GCN | 70.8 | 67.6 | 30.2 | 18.3 | 25.0 | 20.0 |
| | GCN(Drop) | 72.3 | 70.6 | 61.4 | 57.2 | 41.6 | 34.4 |
| | JKNet | - | 68.7 | 67.7 | 69.8 | 68.2 | 63.4 |
| | JKNet(Drop) | - | 72.6 | 71.8 | 72.6 | 70.8 | 72.2 |
| | Incep | - | 69.3 | 68.4 | 70.2 | 68.0 | 67.5 |
| | Incep(Drop) | - | 72.7 | 71.4 | 72.5 | 72.6 | 71.0 |
| | GCNII | 68.2 | 68.9 | 70.6 | 72.9 | 73.4 | 73.4 |
| | GRAND | 75.4 | 74.5 | 74.5 | 73.8 | 73.3 | 71.9 |
| | ACMP | 73.5 | 74.6 | 74.2 | 73.1 | 72.8 | 68.9 |
| | **OGC** (ours) | 70.7 | 74.1 | 76.7 | **77.5** | 77.3 | 76.5 |
| | SGC | 71.9 | 69.5 | 69.9 | 69.9 | 69.4 | 68.7 |
| | S$^2$GC | 71.4 | 72.0 | 72.4 | 73.0 | 72.9 | 72.0 |
| | **GGC** (ours) | 71.9 | 72.5 | 73.5 | 73.8 | 72.9 | 73.0 |
| | **GGCM** (ours) | 72.1 | 73.0 | 73.2 | **73.9** | **73.9** | 73.5 |
| **Pubmed** | GNN-LF/HF | 80.0 | 80.2 | 80.3 | 80.4 | 80.5 | 80.3 |
| | GCN | 79.0 | 76.5 | 61.2 | 40.9 | 22.4 | 35.3 |
| | GCN(Drop) | 79.6 | 79.4 | 78.1 | 78.5 | 77.0 | 61.5 |
| | JKNet | - | 78.0 | 78.1 | 72.6 | 72.4 | 74.5 |
| | JKNet(Drop) | - | 78.7 | 78.7 | 79.1 | 79.2 | 78.9 |
| | Incep | - | 77.7 | 77.9 | 74.9 | **OOM** | **OOM** |
| | Incep(Drop) | - | 79.5 | 78.6 | 79.0 | **OOM** | **OOM** |
| | GCNII | 78.2 | 78.8 | 79.3 | 80.2 | 79.8 | 79.7 |
| | GRAND | 81.3 | 82.7 | 82.4 | 82.2 | 81.3 | 80.5 |
| | ACMP | 79.0 | 79.7 | 79.5 | 79.2 | 79.8 | 78.0 |
| | **OGC** (ours) | 80.7 | 81.6 | 82.5 | **83.5** | 82.5 | 82.1 |
| | SGC | 78.9 | 77.5 | 79.1 | 76.9 | 73.2 | 70.8 |
| | S$^2$GC | 78.6 | 78.6 | 79.4 | 80.0 | 78.5 | 76.8 |
| | **GGC** (ours) | 78.9 | 79.1 | 78.9 | 79.3 | 79.4 | 79.7 |
| | **GGCM** (ours) | 78.8 | 79.3 | 79.9 | **81.5** | 81.1 | 80.3 |

and $v_j$ is always $\frac{\sqrt{D_{ii}+1}}{\sqrt{D_{jj}+1}}$. For example, in SGC and GCN, the ratios of node 11's embeddings and node 4's embeddings are both $\frac{\sqrt{3+1}}{\sqrt{1+1}} = \sqrt{2}$. All these findings are consistent with our Theorem 1, successfully verifying our analysis.

Table 7: Running time (seconds) on Pubmed.

| Type | Method | Running on CPU | | | | | | | Running on GPU | | | | | | |
|---|---|---|---|---|---|---|---|---|---|---|---|---|---|---|---|
| | | Layers/Iterations | | | | | | Total[*] | Layers/Iterations | | | | | | Total[*] |
| | | 2 | 4 | 8 | 16 | 32 | 64 | | 2 | 4 | 8 | 16 | 32 | 64 | |
| supervised | GCNII | 98.02 | 234.60 | 545.44 | 923.61 | 1555.37 | 4729.32 | 8086.36 | 4.60 | 10.76 | 29.64 | 54.08 | 200.20 | 780.24 | 1079.52 |
| | GRAND | 857.23 | 952.29 | 2809.84 | 4361.91 | 6225.39 | 15832.91 | 31039.57 | 35.45 | 70.99 | 274.95 | 2090.65 | 2961.71 | 6829.78 | 12263.54 |
| | OGC | 1.88 | 3.69 | 7.29 | 14.63 | 30.36 | 61.88 | 61.88 | | | | - | | | - |
| unsupervised | SGC | 0.22 | 0.28 | 0.44 | 0.77 | 1.46 | 2.80 | 5.78 | 0.52 | 0.52 | 0.55 | 0.61 | 0.70 | 0.82 | 3.59 |
| | S$^2$GC | 0.25 | 0.36 | 0.56 | 0.94 | 2.16 | 3.66 | 6.88 | 0.57 | 0.60 | 0.62 | 0.66 | 0.75 | 0.92 | 3.99 |
| | GGC | 0.27 | 0.50 | 0.85 | 1.60 | 3.10 | 6.02 | 10.26 | 0.58 | 0.59 | 0.62 | 0.67 | 0.80 | 1.01 | 4.08 |
| | GGCM | 0.37 | 0.57 | 1.06 | 1.93 | 3.77 | 7.27 | 12.45 | 0.60 | 0.63 | 0.64 | 0.69 | 0.85 | 1.09 | 4.35 |

[*] To count the total running time, for the compared supervised deep GCNs (GCNII and GRAND), we sum their running times at different layers, since they have to retrain the model totally once the layer number is fixed. As our method OGC could naturally conduct node classification at each iteration, we adopt its time cost when it has finished the 64-th iteration as the total time. Here, we do not report the running time of OGC on GPU, since this method is actually a shallow method. In all unsupervised methods, we will first get node embedding results and then train a logistic regression classifier (with a single layer neural network) for node classification. Therefore, we remove those repetitive costs in the convolution process, as they both can get embedding results at each iteration.

## C.2 EFFECT OF DEPTHS/ITERATIONS IN VARIOUS METHODS

Table 6 shows the complete classification results w.r.t. various numbers of layers/iterations in some deep GCNs, "no-learning" methods and ours. To fix the over-smoothing problem in those deep models, we adopt Dropedge (Rong et al., 2019) and reuse the results (i.e., GCN(Drop), JKNet(Drop) and Incep(Drop)) reported in Rong et al. (2019) and Chen et al. (2020).

We observe that in most cases, the performance of our methods increases as the iteration goes on. Specifically, our methods all tend to get the best performance at around 16-th or 32-th iteration, and could maintain similar performance as the iteration number increases to 64. These results indicate that by introducing supervised knowledge or preserving graph structure in the convolution process on a graph, we can alleviate the over-smoothing problem and benefit from deep propagation. In contrast, most baselines still heavily suffer from the over-smoothing problem. Of note, once the model depth is fixed, all the compared deep GCNs (like Incep, GCNII and GRAND) need to re-train the entire neural networks and re-tune all hyper-parameters, which would cost lots of time, effort and resource. In contrast and worth noticing, our three methods, which all run in an iterative way, do not have this problem. As a whole, these experimental results demonstrate that all our methods, with significantly fewer parameters, enable deep propagation in a more effective way.

## C.3 EFFICIENCY TESTING

We conduct this test on a laptop with the following configuration: 8 GB RAM, Intel Core i5-8300H CPU, and Nvidia GeForce RTX2060 6 GB GPU. All the codes are written in PyTorch, and all the default hyper-parameters are adopted. For baselines, we adopt two recently proposed deep GCNs (GCNII and GRAND) and two typical no-learning GCNs (SGC and S$^2$GC).

For all methods, we always adopt their default hyper-parameters and settings. Table 7 reports the average running time on Pubmed over 100 runs on both CPU and GPU, w.r.t. different numbers of layers/iterations. First of all, we can clearly see that these two supervised deep GCNs are very costly. In contrast, when running on CPU, our supervised method OGC is at least 130 times and 500 times more efficient than GCNII and GRAND, respectively. Moreover, even when switching to GPU, OGC is still much (around 17 times and 198 times) faster than GCNII and GRAND. This is because: 1) OGC is a shallow method which only involves some sparse matrix multiplications; 2) As a supervised iterative method, OGC can conduct node classification at every iteration naturally. The other observation is that our two unsupervised methods GGC and GGCM have the same efficiency level as SGC and S$^2$GC. The extra time cost in our two methods is due to the IGC operation. We also

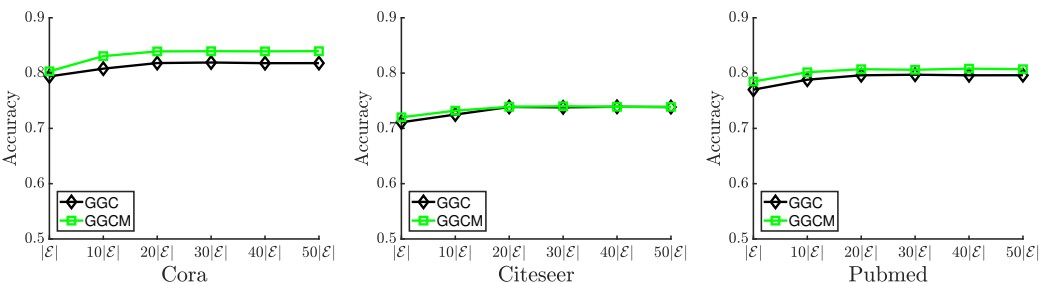

Figure 4: Effect of negative graph edge number in GGC and GGCM.

note that when switching to GPU, our two methods would have very similar efficiency performance as SGC and $S^2$GC.

## C.4    LEARNING WITH BOTH TRAIN AND VALIDATION SETS

| | Cora | | Citeseer | | Pubmed | |
|---|---|---|---|---|---|---|
| | Acc. | Chg. | Acc. | Chg. | Acc. | Chg. |
| LP | 71.5±0.3 | +3.5↑ | 48.9±0.4 | +3.6↑ | 65.8±0.3 | +2.8↑ |
| ManiReg | 62.0±0.3 | +2.5↑ | 63.5±0.3 | +3.4↑ | 73.2±0.3 | +2.5↑ |
| GCN | 70.2±0.3 | -11.3↓ | 61.2±0.1 | -2.8↓ | 66.5±0.2 | -12.5↓ |
| APPNP | 68.7±0.2 | -14.6↓ | 62.1±0.3 | -9.7↓ | 69.6.5±0.4 | -10.5↓ |
| Eigen-GCN | 70.5± 0.5 | -8.4↓ | 62.9±0.5 | -3.6↓ | 71.2±0.3 | -7.4↓ |
| GNN-LF/HF | 69.8± 0.5 | -14.2↓ | 59.8±0.5 | -12.5↓ | 70.6±0.3 | -9.9↓ |
| C&S | 83.6±0.5 | +2.5↑ | 74.5±0.3 | +2.5↑ | 81.1±0.4 | +2.1↑ |
| NDLS | 69.5±0.7 | -15.1↓ | 53.2±0.4 | -20.5↓ | 70.5±0.4 | -10.9↓ |
| ChebNetII | 66.5±0.3 | -17.2↓ | 54.9±0.5 | -17.9↓ | 59.6±0.2 | -20.9↓ |
| OAGS | 68.5±0.6 | -15.4↓ | 58.4±0.7 | -15.3↓ | 62.8±0.7 | -19.1↓ |
| JKNet | 55.1 ± 0.6 | -27.6↓ | 59.4 ± 0.4 | -13.6↓ | 64.2 ± 0.6 | -13.7↓ |
| Incep | 57.5±0.2 | -25.3↓ | 60.2±0.4 | -12.1↓ | 62.5±0.5 | -17.0↓ |
| GCNII | 58.8±0.5 | -26.7↓ | 57.6 ± 0.6 | -15.8↓ | 66.5 ± 0.4 | -13.7↓ |
| GRAND | 56.7±0.4 | -28.7↓ | 52.5±0.4 | -22.9↓ | 59.6±0.6 | -23.1↓ |
| ACMP | 60.3±0.5 | -24.6↓ | 58.4±0.3 | -17.1↓ | 64.5±0.5 | -14.9↓ |
| **OGC** (ours) | **86.9 ± 0.1** | +3.2↑ | **77.5 ± 0.2** | +3.5↑ | **83.4 ± 0.4** | +2.2↑ |

Table 8: Summary of classification accuracy (%) of all supervised methods, with both train and validation label sets are used for model learning. The "Chg." column shows the accuracy change, compared to when only the train part is used for model learning.

In this subsection, for a more fair comparison, we use both training and validation sets as the supervised knowledge for all supervised methods. Specifically, for all shallow baselines and our method OGC, their hyper-parameters are chosen to consistently improve the classification accuracy on the whole label set. For GNN baselines, besides the above-mentioned hyper-parameter tuning strategy, we also try their default settings, and we finally report the best results of these two hyper-parameter tuning strategies. At last, for all baselines, we will stop their learning process when the predicted labels converge.

Table 8 shows the node classification results. We can clearly see that the performance of all shallow methods (including our method OGC) gains significantly. This indicates that shallow methods do not heavily rely on the validation set for hyper-parameter tuning. In contrast, the performance of all GNN baselines (especially those very deep ones) declines dramatically, indicating that they all badly suffer from the overfitting problem. This is consistent with previous deep learning studies which have shown that general GNNs heavily require many additional labels for validation.

## C.5    VARIOUS SETTINGS OF THE PROPOSED METHODS

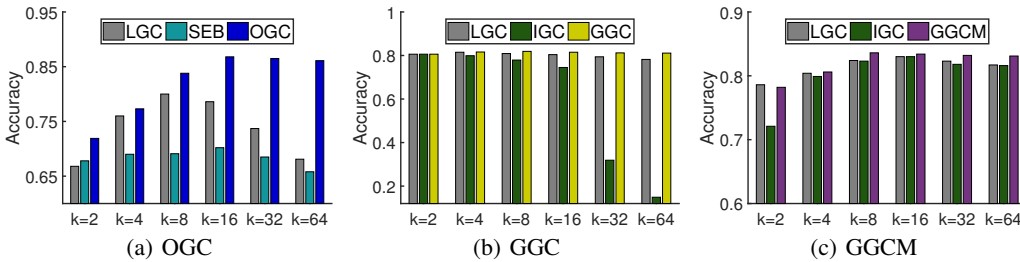

Figure 5: Ablation study on Cora.

### C.5.1 ABLATION STUDY

To completely understand the impact of each model part, we conduct an ablation study by disabling each component in our methods. This experiment is set as follows. In OGC, we test its two sub-parts: lazy graph convolution (short for *LGC*) and SEB. In GGC and GGCM, we also test their two subparts: LGC and IGC. To save space, we only test the first dataset Cora, as the similar results are observed on the other datasets.

As shown in Fig. 5, in all our three methods, separating the training process of their two subparts will lead to worse performance. Moreover, as shown in Fig. 5(b), IGC declines most rapidly, as the iteration number increases. We can intuitively understand this as: for node embedding, merely ensuring the dissimilarity between unlinked nodes is pointless, as those unlinked node pairs are enormous. On the other hand, comparing with the results in Table 6, we also find that the performance of LGC declines much slower than that of SGC. This is consistent with our analysis in Sect. 3.1, that is, lazy graph convolution can be used to alleviate the over-smoothing problem. We also note that the performance of LGC in these three sub-figures looks quite different. This is because our three methods use this operator with different inputs and settings (like $\beta$ values and decay rates).

### C.5.2 SENSITIVITY ON NUMBER OF EDGES IN IGC

In the proposed methods GGC and GGCM, they both randomly generate a negative graph for the IGC operator at each iteration. We vary the edge number in this negative graph among $\{|\mathcal{E}|, 10|\mathcal{E}|, 20|\mathcal{E}|, 30|\mathcal{E}|, 40|\mathcal{E}|, 50|\mathcal{E}|\}$, where $|\mathcal{E}|$ is the edge number of the original graph. For simplicity, we always fix the iteration number to 16. As shown in Fig. 4, at the beginning, when the edge number increases, the performance of these two methods will also increase. However, when the edge number larger than $20|\mathcal{E}|$, the performance of these methods will be very stable. This is consistent with the common sense of negative sampling. For example, in the famous work word2vec (Mikolov et al., 2013b), for each positive pair, it requires 15 negative pairs to get preferred performance. These results indicate that increasing the negative graph edge number within a certain limit could improve the performance of both GGC and GGCM.

### C.5.3 EFFECT OF LIM TRICK IN OGC

In our supervised methods OGC, we adopt the LIM trick. The primary objective behind this adoption is to mitigate the risk of overfitting within the learned node embeddings (i.e., updating $U$ in Eq. 6). To comprehensively assess the impact of this trick, we conduct an ablation study in this subsection. Table C.5.3 shows the classification performance, and the training MSE loss (including both the train and validation sets). We can clearly see that LIM trick largely improve the performance, simultaneously obtaining higher training loss values. This observation suggests that the LIM trick holds promise in addressing the overfitting challenge within graph embedding tasks.

## D DERIVATION DETAILS

|  | Cora | | Citeseer | | Pubmed | |
|---|---|---|---|---|---|---|
|  | Wit. | Wio. | Wit. | Wio. | Wit. | Wio. |
| Accuracy (%) | **86.9** | 84.9 | **77.5** | 75.8 | **83.4** | 82.6 |
| Training MSE Loss | 2.3e-2 | 3.7e-3 | 3.5e-2 | 2.7e-3 | 7.2e-2 | 5.3e-3 |

Table 9: Effect of LIM trick in OGC on node classification. The "Wit." and "Wio." columns show the performance with the LIM trick enabled and disabled, respectively.

### D.1 DERIVATIVE OF GRAPH CONVOLUTION IN SGC

We can minimize the loss $\bar{\mathcal{Q}}_{smo}$ in Eq. 3 by iteratively updating $U$ as:

$$
\begin{aligned}
U^{(k+1)} &= U^{(k)} - \eta_{smo}\frac{\partial \bar{\mathcal{Q}}_{smo}}{\partial U^{(k)}} \\
&= U^{(k)} - 2\eta_{smo}\tilde{D}^{-\frac{1}{2}}\tilde{L}\tilde{D}^{-\frac{1}{2}}U^{(k)},
\end{aligned}
\tag{18}
$$

where $\eta_{smo}$ is the learning rate in this part. If we set $\eta_{smo} = \frac{1}{2}$, we can get:

$$
\begin{aligned}
U^{(k+1)} &= U^{(k)} - \tilde{D}^{-\frac{1}{2}}\tilde{L}\tilde{D}^{-\frac{1}{2}}U^{(k)} \\
&= (I_n - \tilde{D}^{-\frac{1}{2}}\tilde{L}\tilde{D}^{-\frac{1}{2}})U^{(k)} \\
&= (\tilde{D}^{-\frac{1}{2}}\tilde{A}\tilde{D}^{-\frac{1}{2}})U^{(k)}.
\end{aligned}
\tag{19}
$$

Eq. 19 is exactly the graph convolution (i.e., Eq. 2) used in SGC. In addition, SGC initializes the learned node embedding matrix with $X$, which satisfies the constraint in Eq. 3.

### D.2 DERIVATIVE OF LAZY RANDOM WALK IN SGC

We can rewrite Eq. 18 as follows:

$$
\begin{aligned}
U^{(k+1)} &= U^{(k)} - 2\eta_{smo}\tilde{D}^{-\frac{1}{2}}\tilde{L}\tilde{D}^{-\frac{1}{2}}U^{(k)} \\
&= (I_n - 2\eta_{smo}\tilde{D}^{-\frac{1}{2}}\tilde{L}\tilde{D}^{-\frac{1}{2}})U^{(k)} \\
&= [\tilde{D}^{-\frac{1}{2}}(\tilde{D} - 2\eta_{smo}\tilde{L})\tilde{D}^{-\frac{1}{2}}]U^{(k)} \\
&= [\tilde{D}^{-\frac{1}{2}}(\tilde{D} - 2\eta_{smo}(\tilde{D} - \tilde{A}))\tilde{D}^{-\frac{1}{2}}]U^{(k)} \\
&= [2\eta_{smo}\tilde{D}^{-\frac{1}{2}}\tilde{A}\tilde{D}^{-\frac{1}{2}} + (1 - 2\eta_{smo})I_n]U^{(k)}.
\end{aligned}
\tag{20}
$$

Let $\beta = 2\eta_{smo}$ and ensure $\beta \in (0, 1)$. The above equation becomes:

$$
U^{(k+1)} = [\beta\tilde{D}^{-\frac{1}{2}}\tilde{A}\tilde{D}^{-\frac{1}{2}} + (1 - \beta)I_n]U^{(k)}.
\tag{21}
$$

The expression in the square brackets in Eq. 21 is exactly the definition of lazy random walk, and parameter $\beta$ can be seen as the moving probability that a node moves to its neighbors in every period.

### D.3 DERIVATIVE OF CROSS-ENTROPY LOSS

To simplify writing and differentiation, we let $Z = UW$ and $P = \text{softmax}(UW)$. The cross-entropy loss of a single sample $j$ is: $\mathcal{L}_j = -\sum_{k=1}^c Y_{jk}log(P_{jk})$.

First of all, we introduce the derivative of softmax operator:[8]

$$
\frac{\partial P_{jk}}{\partial Z_{ji}} = \frac{\partial \frac{e^{Z_{jk}}}{\sum_{i^*=1}^c e^{Z_{ji^*}}}}{\partial Z_{ji}} = P_{jk}(\delta_{ki} - P_{ji}),
\tag{22}
$$

where $\delta_{ki}$ is the Kronecker delta:

$$
\delta_{ki} = \begin{cases} 1, & \text{if } k = i; \\ 0, & \text{if } k \neq i. \end{cases}
$$

---

[8]https://deepnotes.io/softmax-crossentropy

Then, we can calculate the derivative of $\mathcal{L}_j$ w.r.t. $Z_{ji}$ as follows:

$$\frac{\partial \mathcal{L}_j}{\partial Z_{ji}} = -\sum_{k=1}^{c} Y_{jk} \frac{\partial log(P_{jk})}{\partial P_{jk}} \times \frac{\partial P_{jk}}{\partial Z_{ji}} = P_{ji} - Y_{ji}. \tag{23}$$

After that, the derivative of $\mathcal{L}_j$ w.r.t. $U_{ji}$ can be obtained as follows:

$$\frac{\partial \mathcal{L}_j}{\partial U_{ji}} = \sum_{i^*=1}^{c} \frac{\partial \mathcal{L}_j}{\partial Z_{ji^*}} \times \frac{\partial Z_{ji^*}}{\partial U_{ji}} = \sum_{i^*=1}^{c} (P_{ji^*} - Y_{ji^*}) W_{ii^*}. \tag{24}$$

We use $\mathcal{L}_{all}$ to denote the overall loss, i.e., $\mathcal{L}_{all} = \sum_{j=1}^{n} \mathcal{L}_j$. Combining the derivatives of all the samples together, we can finally get:

$$\frac{\partial \mathcal{L}_{all}}{\partial U} = S(P - Y)W^T, \tag{25}$$

where $S$ is a diagonal matrix with $S_{jj} = 1$ if the simple $j$ is labeled, and $S_{jj} = 0$ otherwise.

### D.4    Derivative of Squared Loss

The squared loss of a single sample $j$ is: $\mathcal{L}_j = \frac{1}{2} \sum_{k=1}^{c} (Y_{jk} - Z_{jk})^2$. Then, we can get the derivative of $\mathcal{L}_j$ w.r.t. $U_{ji}$ as follows:

$$\frac{\partial \mathcal{L}_j}{\partial U_{ji}} = \sum_{i^*=1}^{c} \frac{\partial \mathcal{L}_j}{\partial Z_{ji^*}} \times \frac{\partial Z_{ji^*}}{\partial U_{ji}} = -\sum_{i^*=1}^{c} (Y_{ji^*} - Z_{ji^*}) W_{ii^*}. \tag{26}$$

Considering all the samples together, we can finally get:

$$\frac{\partial \mathcal{L}_{all}}{\partial U} = S(Z - Y)W^T, \tag{27}$$

where $S$ is a diagonal matrix with $S_{jj} = 1$ if the simple $j$ is labeled, and $S_{jj} = 0$ otherwise.

## E    More Comparison Discussion and Related Work

### E.1    Comparison with Existing Work

Our supervised method OGC enjoys high accuracy and low training complexity. High accuracy is due to the ability of utilizing validation labels for model training compared with typical GCN-type methods like GCN (Kipf & Welling, 2017), GAT (Veličković et al., 2018) and GCNII (Chen et al., 2020). Low training complexity is due to the easily parallelizable embedding updating mechanism compared with those end-to-end training GCN-type methods (like N-GCN (Abu-El-Haija et al., 2020) and L$^2$-GCN (You et al., 2020)) that also use label for joint learning.

Our unsupervised methods GGC and GGCM enjoy high graph structure preserving ability, by introducing the IGC operator at each iteration. This makes our methods especially suitable for those graph structure-aware tasks, compared with typical shallow GCN-type methods, like SGC (Wu et al., 2019) and S$^2$GC (Zhu & Koniusz, 2021). Especially, the introduced IGC operator (a high-pass filter) can also be used in the heterophilic graphs, by directly feeding the graph structure into this operator. In addition, compared to some shallow graph embedding methods (like LINE (Tang et al., 2015) and DeepWalk (Perozzi et al., 2014)) whose objective functions are designed to preserve the graph structure, our two methods can better utilize node attributes.

### E.2    Graph-based Semi-Supervised Learning (GSSL)

Generally, GSSL methods can be divided into two categories (Song et al., 2022a). The first are graph regularization based methods, in which a Laplacian regularizer is used to ensure the cluster assumption. For example, Zhou et al. (2004) uses a symmetric normalized Laplacian regularizer; Zhu et al. (2003) adopts a Gaussian-random-field based Laplacian regularizer. The other are graph embedding based methods which aim to learn effective node embeddings (i.e., features) for

node classification (Cai et al., 2018). Prior works, like Singular Value Decomposition (Golub & Reinsch, 1971) and Nonnegative Matrix Factorization (Pauca et al., 2006), mainly rely on graph factorization, badly suffering from the scalability problem. Inspired by the success of word embedding (Mikolov et al., 2013a), some random walk based methods, like DeepWalk (Perozzi et al., 2014) and node2vec (Grover & Leskovec, 2016), are proposed to efficiently capture the connectivity patterns in a graph. Following this, some attributed graph embedding methods (Yang et al., 2015) further consider the node attributes; some supervised methods (Yang et al., 2016) further utilize label information.

### E.3 GRAPH NEURAL NETWORKS (GNNS)

With the power of neural network architectures, graph neural networks (GNNs) (Wu et al., 2020) are recently becoming the primary techniques for graph-structured data learning. Generally, GNNs can also be considered as graph embedding based methods, as the outputs of any layer can be seen as the node embedding results. However, unlike the previous methods, GNNs learn embeddings by aggregating feature information via neural networks at each layer. Early works (Gori et al., 2005; Scarselli et al., 2008) introduce recursive neural networks for feature aggregation. However, these models suffer from scalability issues, as they requires the repeated application of contraction maps until node representations converge. Inspired by the great success of CNNs on grid-structured data, Defferrard et al. (2016) first formally defines the graph convolution operation by aggregating feature in the Fourier domain. After that, GCN (Kipf & Welling, 2017) further simplify the graph convolution operation by using localized first-order approximations of the graph Laplacian matrix. Since then, a large number of GCN variants have been proposed, like Graph Attention Networks (GAT) (Veličković et al., 2018), GraphSAGE (Hamilton et al., 2017), ChebNetII (He et al., 2022) and OAGS (Song et al., 2022b), which have shown state-of-the-art performance on various graph-based tasks. Our methods also follow this line. Additionally, by alleviating the potential suboptimality of GCNs for joint graph structure and supervised learning, our methods affirm the existence of simple but powerful GNN models.

