# OpenReview forum: "From Cluster Assumption to Graph Convolution: Graph-based Semi-Supervised Learning Revisited"
_ICLR.cc/2024/Conference — ICLR 2024 Conference Withdrawn Submission_

### Official Review · Reviewer_seo4 · 2023-10-23

**Soundness:** 2 fair
**Presentation:** 2 fair
**Contribution:** 2 fair
**Rating:** 5
**Confidence:** 3

**Summary:**

The authors propose an approach for graph-based semi-supervised learning through the lens of traditional methods and Graph Convolutional Networks (GCNs). Through an analysis in terms optimization the authors claim that the current approaches emphasize the minimzation of the smoothing term leaving aside the term related to classification accuracy. The authors propose to fix this with a novel update of the nodes embeddings and a further approach based on the complement of a graph.

The authors report experiments showing the effectivity of the proposed approach.

**Strengths:**

The paper provides an interesting view on the behaviour of graph-based models, and how they focus mainly on one term of their objective function. Interestingly, the authors show that by adding information of underlying negative edges do provide an improvement in performance.

**Weaknesses:**

It is unclear how the analysis of embeddings is done. The authors present an analysis where the update of the kth layer is posed as an optimization problem, where the resulting update is in terms of the previous embedding. I wonder why this is the case. I assume that the optimization problem of the current task is on the output provided by the last layer, for which the embedding is provided by the architecture considered by the authors. Instead, the authors propose to analyze the current problem as some sort of nested optimization problems, where each of the embeddings, i.e. $k$, is computed through an optimization problem which is in terms of the previous embedding, i.e. $k-1$. Why is this approach through nested optimization problems suitable? Is it somehow standard in the literature or is it as well a contribution from the authors?

The authors consider the complement of a graph as a way to obtain information that allows the model to pull nodes apart. There are a couple of concerns about this approach:

1. if the original input graph is sparse, then the complement of a graph is dense, which implies a quadratic number of edges in terms of the number of nodes. Given this, to what degree is the proposed approach scalable? In several places the authors briefly mention some sort of sampling, but it is unclear if this is well explored or explained in the paper.

2. the notion of a complement of a graph in the context of this paper is related to negative edges, i.e. edges that encode conflict, enmity or polarity, and hence they are useful to pull nodes apart in the embedding. This kind of edges have been explored previously in the literature in the context of spectral clustering and graph-based semi-supervised learning. Examples of previous works are:

    a. Node classification for signed networks using diffuse interface methods, 2019

    b. A Linearly Constrained Power Iteration for Spectral Semi-Supervised Classification on Signed Graphs, 2022

    c. Dissimilarity in graph-based semisupervised classification, 2007

    d. Node classification in signed social networks, 2016

**Questions:**

1. Can the authors provide an insight, motivation or design choice for analyzing the current approach as a set of nested optimization problems rather than directly studying the original optimization problem, which is based on the embedding of the last layer?
2. Can the authors elaborate on how the proposed approach handles the large amount of edges that are potentially observed in the complement of a graph?

---

### Official Review · Reviewer_WdoQ · 2023-10-30

**Soundness:** 2 fair
**Presentation:** 3 good
**Contribution:** 2 fair
**Rating:** 5
**Confidence:** 4

**Summary:**

This paper explores the relationship between traditional shallow learners and graph convolutional networks (GCNs) in the context of graph-based semi-supervised learning (GSSL). The study reveals a crucial distinction where typical GCNs may not effectively integrate both graph structure and label information at each layer. To address this, the authors propose three graph convolution methods: a supervised approach called OGC that leverages label information, and two "no-learning" unsupervised methods, GGC and its multi-scale version GGCM, designed to preserve graph structure information during convolution. Extensive experiments demonstrate the effectiveness of these proposed methods.

**Strengths:**

The paper introduces three new graph convolution methods, addressing the limitation of typical GCNs in effectively incorporating both graph structure and label information at each layer. These methods offer innovative solutions to improve the performance of GSSL tasks. The authors conduct extensive experiments to empirically demonstrate the effectiveness of their proposed graph convolution methods. These experiments provide practical evidence of the benefits of their approaches in real-world applications

**Weaknesses:**

I appreciate the effort put into incorporating label information during the propagation process at each layer. However, I find the motivation for doing so somewhat unclear. The authors repeatedly assert that the GCN solution solely optimizes the L_smooth, which is indeed true when considering the equivalent of message passing. However, in actual training, GCN still utilizes a supervised loss to train the network, and label errors propagate through all layers. Therefore, the advantages of integrating label information into the proposed framework require further explanation and analysis to clarify their significance.

**Questions:**

See weakness

---

### Official Review · Reviewer_9Ae7 · 2023-11-03

**Soundness:** 2 fair
**Presentation:** 2 fair
**Contribution:** 1 poor
**Rating:** 3
**Confidence:** 4

**Summary:**

This paper revisits the connection between message passing based GNN (GCN and SGC) and graph regularized semi supervised learning optimization. The connection motivates the author to design three "new" graph convolution methods: one supervised and two unsupervised way. The author did experiments over old benchmark datasets.

**Strengths:**

1. The author analyzes the connection between GCN and graph regularized optimization in depth, the content is comprehensive.

**Weaknesses:**

1. The content is not new. There were many studies around the topic and the connection to graph-regularized optimization back to 2020. I don't think there is any new discovery out from existing literature. This also means that the author didn't have done a comprehensive literature review in this topic. Since I have not pay attention to this area for years, I can name one that is highly related: ICML workshop 2020, "Connecting Graph Convolutional Networks and Graph-Regularized PCA". There were also some works like unrolling the optimization process to build a graph convolution operator, which are also highly related to methods designed in this paper.

2. The topic is kind of out-of-date for ICLR conference. This topic is widely studied before and I personally don't find it is interesting or meaningful to revisit this topic.

3. The experimental part is also week. Revisiting with existing old benchmark datasets is not valuable. The author at least should consider modern benchmarks like OGB.

**Questions:**

I personally probably don't have enough interest over the topic shown in this paper. Hence I won't give questions. But I'm looking forward to see whether there is some interesting discussion by other reviewers.

---

### Official Review · Reviewer_2zfL · 2023-11-10

**Soundness:** 2 fair
**Presentation:** 1 poor
**Contribution:** 1 poor
**Rating:** 3
**Confidence:** 4

**Summary:**

Summary: The authors analyze Graph Convolutional Networks (GCN and SGC) with an optimization perspective of minimizing the label prediction and embedding smoothing loss. They point out that these two methods do not optimize for label prediction at every layer, and to overcome this, they propose a new model that addresses this issue. In addition to this, they also present two unsupervised GNN versions.

**Strengths:**

- The proposed method achieves significant performance improvement in their experiments.

**Weaknesses:**

Weakness:
- I believe the issue pointed out is commonly known and it might not be an issue in the first place with some simple tricks. The gradients from the label loss influence the representation in the earlier layers and also, having dense layer connections from every layer to the last prediction layer is a simple solution to the problem.
- Many sophisticated GNN variants now handle label information [see references below] better. The authors' analyses are restricted to simpler models alone.
- While the results on small-scale datasets are promising, they need support from larger datasets like the OGB benchmark. The experiments need to compare all methods that use label information. I noticed that C&S results are missing on the reddit dataset.
- Also, the assumption enforced is label smoothness and not the main cluster smoothness assumption.
References:
* Scaling Graph Propagation Kernels for Predictive Learning
- COMBINING LABEL PROPAGATION AND SIMPLE MODELS OUT-PERFORMS GRAPH NEURAL NETWORKS
* Structure-Aware Label Smoothing for Graph Neural Networks

**Questions:**

Check weaknesses above